# The climate opportunities and risks of contrail avoidance

Jessie R. Smith [1] ✉, Carla Grobler [2], Paul J. Hodgson[1], Jayant Mukhopadhaya [3], Marc L. Shapiro[4], Matteo Mirolo[4,5], Marc E. J. Stettler [6], Sebastian D. Eastham [7] & Steven R. H. Barrett [1]

Navigational contrail avoidance presents an opportunity for rapid reduction in aviation-attributable warming. Here, we use the Aviation Climate and Air Quality Impacts model to evaluate the global temperature changes associated with contrail avoidance towards 2050. If no avoidance is adopted, aviation is projected to contribute 0.040 K of $CO_2$ warming and 0.054 K of contrail warming by 2050. The combined warming from aviation $CO_2$ and contrails is 19% of the difference between current temperatures and the +2 °C limit above pre-Industrial levels, i.e. 19% of our remaining temperature budget. An avoidance strategy phased in over 2035-2045 may recover 9% of this budget, but a 10-year delay may reduce this to 2%. The warming due to additional $CO_2$ emitted during avoidance is two orders of magnitude lower than the expected contrail warming reduction. For every year of delay, the world will be on average 0.003 K hotter in 2050. The most significant climate risk associated with contrail avoidance is therefore inaction.

Condensation trails ("contrails") are clouds of ice crystals that form in the exhaust of an aircraft which, like carbon dioxide ($CO_2$) emissions, contribute to global anthropogenic warming[1,2]. The current warming due to contrails is similar to that of all accumulated aviation $CO_2$ emissions since the beginning of the jet age[3–5]. The warming behaviour of contrails however is fundamentally different to that of $CO_2$, in terms of instantaneous strength and lifetime of impact[6,7]. This means that, while fossil $CO_2$ is the leading cause of anthropogenic warming to date[8], contrail mitigation presents an opportunity for warming reduction that is both similar in magnitude and faster in response than almost any $CO_2$ mitigation measure.

This study focuses on quantifying the size and speed of the reduction in global surface temperature rise that is achievable through contrail avoidance. Navigational contrail avoidance (herein referred to as "contrail avoidance") involves the deviation of flight paths to avoid the regions of the atmosphere where persistent contrails are formed (known as ice supersaturated regions, or ISSRs)[9,10].

Contrail avoidance is currently undergoing tests in operational environments[11,12], suggesting that its implementation could be faster and cheaper than other warming mitigation measures of lower maturity (e.g. use of alternative fuels)[13,14]. While contrail avoidance leads to a reduction in contrail warming, there can be a small increase in (<2% on a fleet-wide average[10]) fuel burn and so a $CO_2$ warming penalty associated with the deviation of aircraft from hypothetical fuel-optimal routings. The relative sizes of contrail warming reduction and $CO_2$ warming increase thus determine the climate impact of contrail avoidance[15].

We here define "effectiveness" as the reduction in global contrail effective radiative forcing due to a fleet-wide mitigation measure. Effective radiative forcing (ERF) is a measure of the climate forcing due to an emission after allowing for tropospheric temperature adjustments. At present, the prediction of the geolocation and vertical extent of ISSRs is limited. This in turn imposes a limit on the effectiveness of contrail avoidance since a planned manoeuvre may be a) insufficiently

[1]Department of Engineering, University of Cambridge, Trumpington Street, Cambridge, UK. [2]Department of Mechanical and Aeronautical Engineering, University of Pretoria, Hatfield, Pretoria, South Africa. [3]International Council on Clean Transportation, 1500 K Street NW, Suite 650, Washington, DC, USA. [4]Breakthrough Energy, Kirkland, Washington, USA. [5]Cambridge Institute for Sustainability Leadership, The Entopia Building 1 Regent Street, Cambridge, UK. [6]Department of Civil and Environmental Engineering, Imperial College London, London, USA. [7]Brahmal Vasudevan Institute for Sustainable Aviation, Department of Aeronautics, Imperial College London, London, USA. ✉e-mail: jrs201@cam.ac.uk

large to avoid the full extent of the ISSR, b) unneeded since an ISSR was incorrectly predicted, or c) perverse since the manoeuvre deviates into an ISSR rather than away from it[16,17]. There are other examples of factors that can impact the effectiveness of contrail avoidance, such as conflicts between the flight paths of deviating and non-deviating aircraft and limitations on air traffic controller capacity[9]. It is possible to estimate effectiveness of contrail avoidance from flight trial and ISSR forecast data[11,12,16,18] as in Smith et al.[19]. However, there is high uncertainty in this estimate, and there are likely to be technological and operational advancements as further flight trials are conducted. It is therefore prudent to examine a wide range of effectiveness values to fully explore the climate impacts of contrail avoidance.

Contrail avoidance is not the only near-term aviation climate mitigation measure that is under consideration. Alternative fuels, either derived from biomass or synthetically produced, can be modified to have similar chemical and physical properties to fossil fuel-derived Jet A (the form of kerosene currently used throughout most of the aviation fleet)[20]. Alternative fuels, when paired with appropriate environmental safeguards, have lower lifecycle $CO_2$ emissions than Jet A, so their usage is mandated, in some regions, to increase towards 2050[21]. Such fuels, with their modified composition, have been presented as an alternative means of contrail warming mitigation as they reduce soot emissions[22,23]. The effectiveness of this is projected to be 42% (range: −18% to 81%)[24]. While contrail mitigation through modified fuel composition looks promising, this study focuses on contrail avoidance as a more likely near-term opportunity for contrail management[22].

Previous works relevant to contrail warming mitigation primarily focus on either the climate impact of contrails, or the effectiveness of contrail mitigation measures[25,26]. Various studies explore the climate impact of contrails in terms of their ERF. Lee et al.[3] compares recent ERF from historical aviation emissions, including $CO_2$, contrails, and other non-$CO_2$ effects, while Bock and Burkhardt[27] focuses on estimating the ERF of contrails into the future. Aamaas et al.[5] and Klöwer et al.[4] instead analyse the global surface temperature change due aviation $CO_2$ and non-$CO_2$ under various future aviation emission scenarios. Bickel et al.[28] concludes that the ERF impact from contrails likely has a lower global surface temperature impact than the same ERF from $CO_2$. While the ERF reduction due to technological change and

the use alternative fuels are accounted for, contrail avoidance is not included in these works.

Conversely, Dray et al.[24] estimates the ERF reduction of modified fuel composition and a single contrail avoidance strategy with an assumed effectiveness and fuel burn penalty. Grewe et al.[14], Meuser et al.[29] and Simorgh et al.[30] analyse the impact of climate optimal routing, which includes contrail avoidance alongside minimising $CO_2$ and non-$CO_2$ emissions such as nitrogen oxides. Frias et al.[10], Teoh et al.[9], and Smith et al.[19] explore the ERF reduction and fuel burn penalty associated with various contrail avoidance strategies. Schumann et al.[31] investigates the global surface temperature rise due to a contrail avoidance strategy, but remarks that the work to integrate of the presented results into a climate model is ongoing. Critically, no existing literature is known to use climate modelling to analyse the climate impacts of contrail avoidance at various levels of effectiveness, dates of first action, and fuel burn penalties, so as to comprehensively investigate the opportunities and risks that contrail avoidance presents.

In this study, the Aviation Climate and Air quality Impacts (ACAI) model is used to quantify the temperature impact of fleet wide contrail avoidance. After a preliminary exploration of various aviation growth projections towards 2100, the presented analysis investigates:

1) The "no action" scenario: the $CO_2$ and contrail warming impacts of aviation in a scenario where there is 3% growth in flown kilometres per year and no contrail avoidance action is taken.
2) The contrail warming reduction of various contrail avoidance scenarios, i.e. the reduction in the surface temperature rise due to contrails when avoidance is adopted relative to the "no action" scenario.
3) The magnitude of contrail avoidance temperature increase due to fuel burn, relative to its contrail warming reduction.

In this way, the opportunities and risks of early contrail avoidance action are evaluated in detail.

## Results and discussion
### Aviation growth projections towards 2100
Figure 1 explores the global surface temperature rise due to contrails in the case where no contrail action is taken. The profile is given between

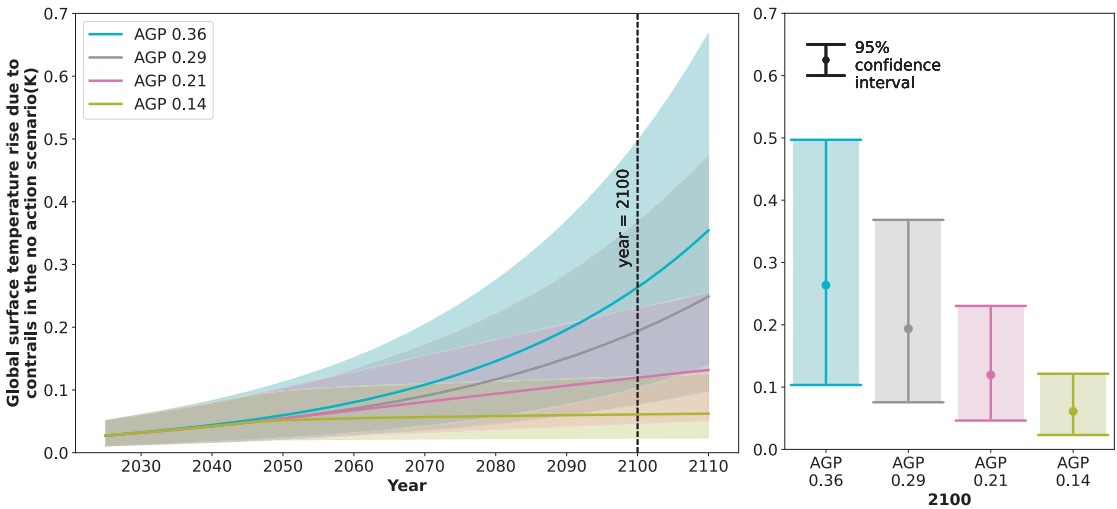

**Fig. 1 | The global surface temperature rise due to contrails for various future scenarios related to flown distance and contrail forcing, where no contrail avoidance action is taken.** The distribution is presented from 2025 to 2110 (left) and for 2100 (right). The mean response (solid line) and 95% confidence intervals (shaded area) are depicted for various future aviation growth projections (AGPs) that relate to flown distance and contrail forcing. These projections are as follows: exponential growth in flown distance after 2045 where saturation in future contrail forcing effects is not assumed (AGP 0.36) and is assumed (AGP 0.29); and linear growth in flown distance after 2045 (AGP 0.21) and no growth in flown distance after 2045 (AGP 0.14), where saturation in future contrail forcing effects is assumed.

**Table 1 | The global surface temperature rise due to aviation in 2050 and 2100, in the case where no action is taken**

| Aviation growth projection (AGP) | Global surface temperature rise (Mean [2.5% CI, 97.5% CI] K) | |
| --- | --- | --- |
| | In 2050 | In 2100 |
| (A) Due to contrails (see also Fig. 1) | | |
| AGP 0.36 | 0.060 [0.023, 0.112] | 0.264 [0.103, 0.497] |
| AGP 0.29 | 0.054 [0.021, 0.103] | 0.194 [0.076, 0.368] |
| AGP 0.21 | 0.054 [0.021, 0.103] | 0.120 [0.046,0.230] |
| AGP 0.14 | 0.052 [0.020, 0.098] | 0.061 [0.024, 0.121] |
| (B) Due to aviation carbon dioxide (see also Supplementary Fig. 1 II) | | |
| AGP 0.36 | 0.040 [0.025, 0.061] | 0.091 [0.056, 0.142] |
| AGP 0.29 | 0.040 [0.025, 0.061] | 0.091 [0.056, 0.142] |
| AGP 0.21 | 0.040 [0.025, 0.061] | 0.086 [0.052, 0.135] |
| AGP 0.14 | 0.040 [0.025, 0.061] | 0.080 [0.049, 0.127] |

A summary of the four aviation growth projections presented in Fig. 1 and Supplementary Fig. 1 II.

the years 2025 (present day) and 2110, with the year 2100 highlighted, and the distribution in global surface temperature rise due to contrails (i.e. mean and 95% confidence intervals) are also depicted. Four future aviation growth projections (AGPs) are presented, labelled AGP 0.36, AGP 0.29, AGP 0.21 and AGP 0.14. The number proceeding "AGP" in the label indicates the mean global surface temperature rise in kelvin due to contrails and aviation related carbon dioxide emissions in the case where no contrail avoidance action is taken. Table 1 summarises the values in Fig. 1 for the years 2050 and 2100[31].

Up to 2045, a 3% growth in flown kilometres per year is assumed (as projected by the International Council on Clean Transportation[32]). There is little literature data on aviation demand growth beyond 2045, so the AGPs have been constructed to cover a range of possible growth rates in annual kilometres flown and annual $CO_2$ emissions towards 2100. The projections AGP 0.36 and AGP 0.29 continue the 3% growth in flown kilometres per year towards 2100, whereas AGP 0.21 and AGP 0.14 assume a linear increase and no increase in flown kilometres beyond 2045, respectively.

The AGPs also explore the relationships between annual kilometres flown and contrail ERF that are present in the literature. While AGP 0.36 assumes a linear relationship between annual kilometres flown and contrail ERF as in Lee et al.[3], the AGP 0.29, AGP 0.21 and AGP 0.14 assume that contrail ERF is related to kilometres flown via a power relationship so as to align with the growth in contrail ERF presented in Bock and Burkhardt[27]. In these scenarios, the contrail ERF is seen to "saturate" in Fig. 1 i.e. it increases less-than-linearly with increases in annual kilometres flown.

It can also be seen in Fig. 1 and Table 1 that there is a slight but noticeable difference between the four scenarios in 2050, and a more pronounced difference between the scenarios by 2100. In 2100, there is a 0.2 K difference between the means of the AGPs with the highest and lowest temperature rise, whereas in 2050, the difference between AGPs with the highest and lowest temperature rise is 0.008 K.

The use of different AGPs therefore leads to noticeable differences in the results, but the differences are sufficiently small to leave the conclusions of this work unchanged. Throughout the rest of this analysis, the opportunities associated with contrail avoidance towards 2050 are explored in The no action scenario and The impact of different contrail avoidance scenarios sections using the AGP 0.29. The risks associated with contrail avoidance in 2050 and 2100 are explored in The risk associated with contrail avoidance section using AGP 0.14. For completeness, each contrail AGP is presented in extended versions of Fig. 2 to Fig. 7 in the Supplementary information. The global surface temperature rise due to aviation $CO_2$ is also altered by the four growth scenarios, as seen in Table 1 and in Supplementary Fig. 1 in the Supplementary information.

## The no action scenario

Figure 2 presents the global surface temperature rise due to aviation $CO_2$ emissions and contrails in the scenario where no contrail avoidance is attempted (i.e. the "no action" scenario). Here, the profile is given between the years 2025 and 2055, with the year 2050 highlighted. The distribution in global surface temperature rise (i.e. mean and 95% confidence intervals) is depicted as an addition to mean $CO_2$ warming. The aviation growth projection AGP 0.29 is here assumed.

The mean global surface temperature rise due to contrails is 0.027 K in 2025, and 0.054 K in 2050. These values are 115% of the mean aviation $CO_2$ warming in 2025 (0.024 K) and 136% of the mean aviation $CO_2$ warming in 2050 (0.040 K). In this contrail ERF growth scenario therefore, the severity of the contrail climate impact is projected to increase relative to that of aviation $CO_2$ if no contrail avoidance action is taken.

The 2015 Paris Agreement aims to limit global surface temperature rise to less than 2 K above pre-industrial levels[26,33]. The current global surface temperature rise above pre-industrial levels is approximately 1.4 K[34]. If all anthropogenic emissions were to cease instantly, the IPCC predicts that global surface temperatures would peak at approximately 0.1 K above the temperature at that time (Figure 1.5 of the Global Warming of 1.5 °C Special Report[35]). The committed temperature rise above industrial levels is therefore estimated to be 1.5 K, which means that the remaining global surface temperature budget is 0.5 K. Beyond this, the 2 K Paris Agreement limit would be exceeded.

Hypothetically, if all contrail warming was eliminated by 2050 (accounting for both the implementation of contrail avoidance and the atmospheric-ocean inertia in global surface temperature rise), this would be a reduction in global surface temperature rise equivalent to 11% of the remaining global temperature budget. This demonstrates the opportunity for warming reduction that contrail avoidance presents.

## The impact of different contrail avoidance scenarios

Figures 3, 4, 5 summarise the effect that phased contrail avoidance implementation has on global surface temperature rise. As in Fig. 1, the contrail warming impact is presented in isolation, without the warming impact of aviation $CO_2$ emissions. For 2025 to 2055, the warming impact of various contrail avoidance strategies is depicted. Over the span of 10 years, all strategies are shown to progress from no adoption to fleet-wide adoption. Different levels of contrail avoidance effectiveness (i.e. the reduction in global contrail ERF due to a fleet-wide contrail avoidance scenario), different starting years, and the impact of contrail avoidance and modified fuel composition are explored. Throughout this section, the aviation growth projection AGP 0.29 is assumed. For each scenario in Fig. 3, Fig. 4 and Fig. 5, the reduction in

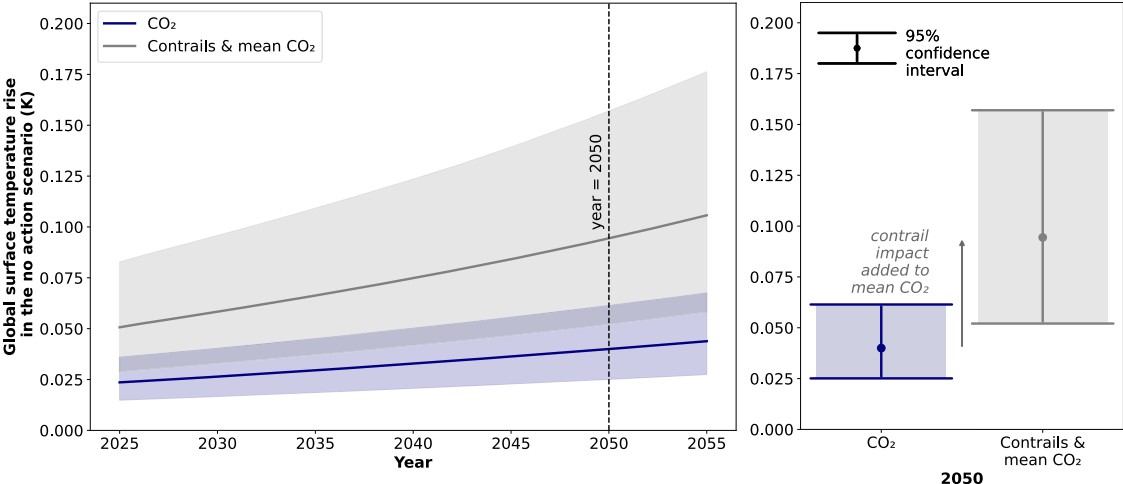

**Fig. 2 | The global surface temperature rise in the scenario where no contrail avoidance action is taken.** The distribution is presented from 2025 to 2055 (left) and for 2050 (right). The mean response (solid line) and 95% confidence intervals (shaded area) for contrails are depicted as an addition to the mean response for aviation $CO_2$ emissions. The aviation growth projection (AGP) AGP 0.29 is assumed, i.e. exponential growth in flown distance beyond 2045 and a saturation in future contrail forcing effects.

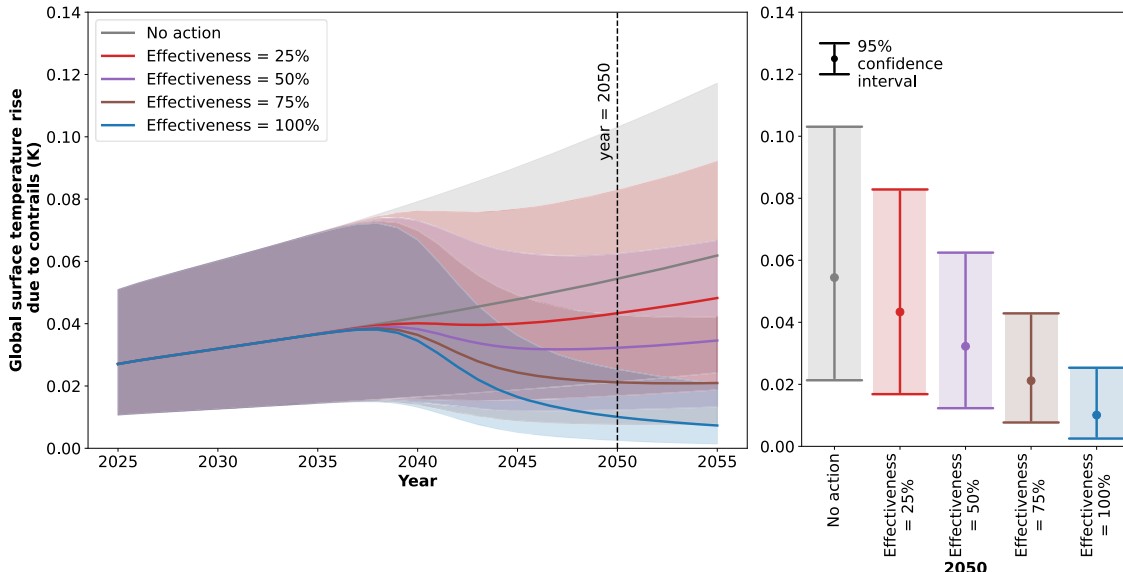

**Fig. 3 | The global surface temperature rise due to contrails, for contrail avoidance scenarios with different levels of effectiveness.** The distribution is presented from 2025 to 2055 (left) and for 2050 (right). The mean response (solid line) and 95% confidence intervals (shaded area) are depicted for contrail avoidance scenarios with levels of contrail avoidance effectiveness of 0% (here equivalent to no contrail avoidance action, i.e. "No action"), 25%, 50%, 75% and 100%. All scenarios have a 2035 start date, and progress from no adoption to fleet wide adoption over the course of 10 years. The aviation growth projection (AGP) AGP 0.29 is assumed, i.e. exponential growth in flown distance beyond 2045 and a saturation in future contrail forcing effects.

global surface temperature rise in 2050 relative to the "no action" scenario is also summarised in Table 2.

Figure 3 presents contrail avoidance scenarios with a start date of 2035, and 5 different levels of contrail avoidance effectiveness (0%– equivalent to the "no action" scenario, 25%, 50%, 75% and 100%). If contrail avoidance is 100% effective, a mean reduction in global surface temperature rise of 0.044 K is achieved by 2050, relative to the scenario where no contrail avoidance action is taken. This reduction can be equated to 9% of the global temperature budget that remains within the Paris Agreement to limit warming to +2 °C above pre-industrial levels[33] (see also Table 2). To two significant figures, the mean reduction in global surface temperature rise due to contrail avoidance is seen to scale linearly with effectiveness.

It is unlikely that contrail avoidance will be 100% effective from the first date of action. However, a 100% effectiveness is used here to get an upper bound estimate of the opportunity loss and climate damage of delayed action. In Fig. 4, contrail avoidance scenarios with 100% contrail avoidance effectiveness, and start times of 2035, 2040 and 2045 are presented. If the adoption of contrail avoidance begins in 2040 instead of 2035, a mean reduction in global surface temperature rise of 0.036 K is achieved by 2050, whereas if it begins in 2045, this reduction decreases to 0.010 K.

It is therefore evident that delaying contrail avoidance action by 10 years causes climate damage equivalent to 0.034 K of warming by 2050, or 7% of the remaining global temperature budget. Since the reduction in global surface temperature rise due to contrail avoidance

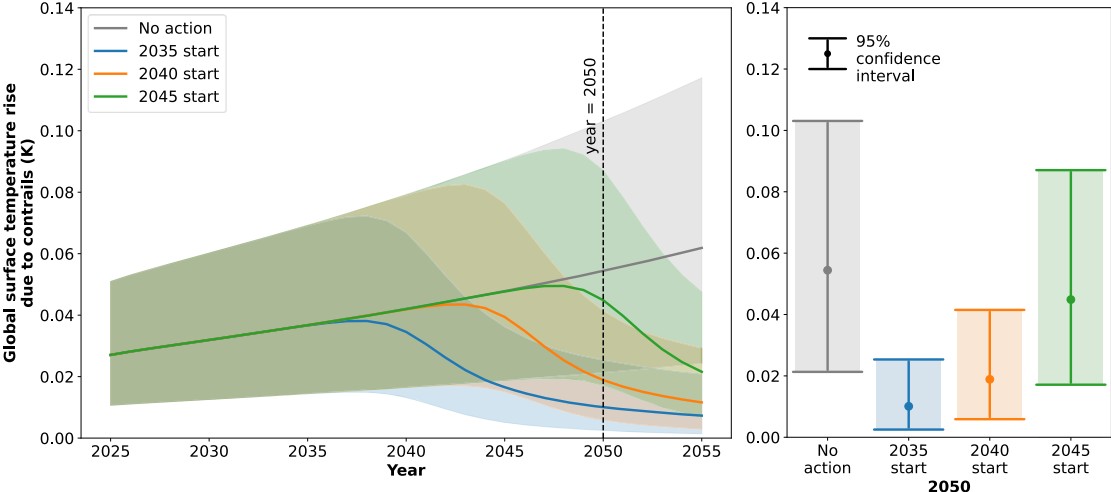

**Fig. 4 | The global surface temperature rise due to contrails, for contrail avoidance scenarios with different start dates.** The distribution is presented from 2025 to 2055 (left) and for 2050 (right). The mean response (solid line) and 95% confidence intervals (shaded area) are depicted for contrail avoidance scenarios with start times of 2035, 2040, and 2045. All scenarios have 100% contrail avoidance effectiveness, and progress from no adoption to fleet wide adoption over the course of 10 years. The scenario where no contrail avoidance action is taken (No action) is also presented. The aviation growth projection (AGP) AGP 0.29 is assumed, i.e. exponential growth in flown distance beyond 2045 and a saturation in future contrail forcing effects.

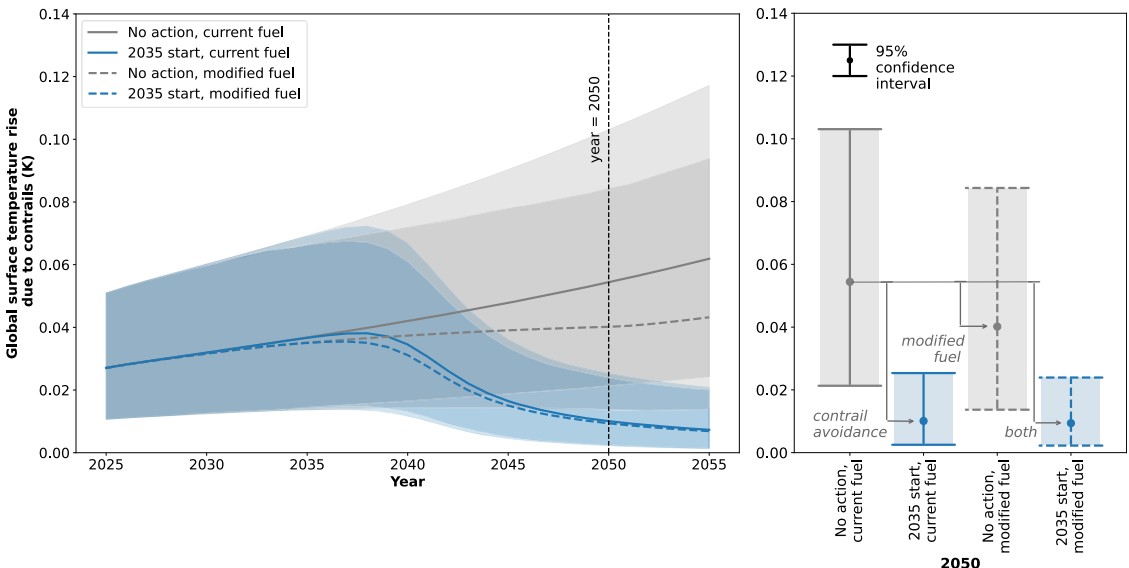

**Fig. 5 | The global surface temperature rise due to contrails, for contrail avoidance scenarios with and without the modification of fleet-wide fuel composition.** The distribution is presented from 2025 to 2055 (left) and for 2050 (right). The mean response with (dashed line) and without (solid line) modified fuel composition and 95% confidence intervals (shaded area) are depicted. All scenarios have a 2035 start date, have a 100% contrail avoidance effectiveness, and progress from no adoption to fleet-wide adoption over the course of 10 years. The scenario where no contrail avoidance action is taken (No action) is also presented. The aviation growth projection AGP (AGP) 0.29 is assumed, i.e. exponential growth in flown distance beyond 2045 and a saturation in future contrail forcing effects.

in 2050 scales approximately linearly with effectiveness, the contrail avoidance scenario with 100% effectiveness and a start date of 2045 has approximately the same reduction in global surface temperature rise as the scenario with 22% effectiveness and a start date of 2035. It follows that a delay of 10 years is approximately equivalent to a 78% loss in effectiveness.

Further, the global surface temperature rise due to contrails has a lower maximum value and declines earlier for earlier dates of first action. In the scenario where contrail avoidance is first introduced in 2035, the mean temperature peaks at 0.038 K in 2038. In the scenarios where it is first introduced in 2040 and 2045, the mean temperature peaks at 0.043 K in 2043 and 0.049 K in 2048, respectively. It follows that the reduction in global surface temperature rise integrated over

time to 2050 (a measure of the climate damage that has been prevented to 2050[2]) is higher in value for earlier start dates, equal to a mean of 0.325 K year for 2035, 0.144 K year for 2040, and 0.018 K year for 2045.

These observations highlight the importance of early contrail avoidance action, even if the operational and technological systems associated with the contrail avoidance strategy are not fully mature. Within realistic bounds, a contrail avoidance strategy with a start date of 2035 can lead to an earlier reduction, a lower peak, and a greater reduction in the global surface temperature rise than a strategy where action is delayed for the sake of higher technological readiness. Further, earlier contrail avoidance action provides more opportunities for learning as testing, development and the adoption of best practices

**Table 2 | The reduction in global surface temperature rise due to contrails in 2050**

| | Absolute reduction in global surface temperature rise (Mean [2.5% CI, 97.5% CI] K) | Of the remaining temperature budget (Mean [2.5% CI, 97.5% CI] %) |
|---|---|---|
| (A) Effectiveness, see also Fig. 3<br>Defaults: Start Date = 2035; Fuel = Current fuel and avoidance | | |
| 0% | 0.000 [0.000, 0.000] | 0.0 [0.0, 0.0] |
| 25% | 0.011 [0.004, 0.020] | 2.2 [0.9, 4.0] |
| 50% | 0.022 [0.009, 0.041] | 4.4 [1.8, 8.1] |
| 75% | 0.033 [0.014, 0.060] | 6.7 [2.7, 12.0] |
| 100% | 0.044 [0.019, 0.078] | 8.9 [3.8, 15.5] |
| (B) Start Date, see also Fig. 4<br>Defaults: Effectiveness = 100%; Fuel = Current fuel and avoidance | | |
| No action | 0.000 [0.000, 0.000] | 0.0 [0.0, 0.0] |
| 2035 | 0.044 [0.019, 0.078] | 8.9 [3.8, 15.5] |
| 2040 | 0.036 [0.015, 0.062] | 7.1 [3.1, 12.3] |
| 2045 | 0.010 [0.004, 0.016] | 1.9 [0.8, 3.2] |
| (C) Fuel, see also Fig. 5<br>Defaults: Start Date = 2035; Effectiveness = 100% | | |
| Current fuel, no avoidance | 0.000 [0.000, 0.000] | 0.0 [0.0, 0.0] |
| Current fuel and avoidance | 0.044 [0.019, 0.078] | 8.9 [3.8, 15.5] |
| Modified fuel, no avoidance | 0.014 [0.008, 0.019] | 2.9 [1.5, 3.7] |
| Modified fuel and avoidance | 0.045 [0.019, 0.079] | 9.0 [3.8, 15.8] |

A summary of the various scenarios presented in Figs. 3, 4, 5.

occur. This means that an earlier start date is likely to increase the effectiveness by 2050, amplifying the warming reduction that can be achieved by this date.

In Fig. 5 the impact of modified fuel composition, contrail avoidance (at 100% effectiveness) and the combination of both mitigation measures on global surface temperature rise is presented. Due to lower soot emissions from the exhaust of the aircraft, it is assumed that a modified fuel composition results in an assumed 42% reduction in contrail ERF (the mean reduction estimated in Dray et al.[24]). The modified fuel composition is first introduced in 2025, and it fully replaces the conventional fuel in the fleet by 2050.

It is evident from Fig. 5 that the modified fuel composition can achieve a mean reduction in global surface temperature rise due to contrails of 0.014 K or 3% of the remaining temperature budget in 2050. This reduction is approximately equivalent to a contrail avoidance scenario with an effectiveness of 32% and a start date of 2035. If fuel modifications are instead introduced to the fleet alongside a contrail avoidance strategy, a mean additional temperature reduction of only 0.0007 K is achieved, equivalent to 0.14% of the remaining temperature budget, or an approximate 1.6% increase in contrail avoidance effectiveness. Conversely, the mean additional temperature reduction due to contrail avoidance in the scenario where the fleet employs a modified fuel composition is 0.031 K, equivalent to 6% of the remaining temperature budget, or an approximate 69% increase in contrail avoidance effectiveness.

Therefore, a contrail avoidance scenario which is first implemented in 2035 is expected to achieve a greater contrail warming reduction than modified fuel compositions, provided the effectiveness of contrail avoidance is greater than approximately 32%. This assertion neglects any improvements in effectiveness that may occur as contrail avoidance is adopted throughout the fleet. In such cases, contrail avoidance achieves a greater contrail warming reduction

than modified fuel compositions at effectiveness values that are <32%.

## The risk associated with contrail avoidance

Figure 6 presents the distribution in reduction of contrail-related global surface temperature rise due to contrail avoidance, and the corresponding temperature increase due to increased $CO_2$, for the years 2050 and 2100. The presented case has been selected to represent a pessimistic contrail avoidance scenario: it assumes the aviation growth projection AGP 0.14 and a contrail avoidance effectiveness of 25%. The results are presented as both box plots and scatter plots, where each point represents one of the Monte Carlo simulations in the Monte Carlo set. Here, the first date of implementation is assumed to be 2035 and the time from first action to full-scale adoption is 10 years. Three levels of fuel burn penalty are investigated at full adoption: 0.35%, 5% and 10% of the fleet averaged fuel consumption. Of the three fuel burn penalties, 0.35% is the closest to the distribution of values evidenced in recent literature[10,36,37], and 5% and 10% fuel burn penalty represent two different levels of pessimism for heuristic purposes[31,38–43].

In Fig. 6A, C, the dashed lines indicate the level of contrail warming reduction that is equal to the increase in $CO_2$ warming due to the fuel burn penalty. In the case of an extreme 10% fuel burn penalty at full adoption, the warming eliminated by contrail avoidance at the 2.5% percentile is greater than or equal to the 97.5% percentile of the additional $CO_2$ warming in all 10,000 simulated cases in 2050—i.e. there were no cases where the additional warming is greater than the warming eliminated by contrail avoidance. In 2100, the warming eliminated by contrail avoidance is greater than or equal to the additional $CO_2$ warming in 9096 out of all 10,000 simulated cases—i.e. the reduction in contrail warming is predicted to be greater than the $CO_2$ warming penalty in 90.96% of cases.

It can be seen in Fig. 6B, D that, even in the worst-case scenario (i.e. 10% fuel burn penalty at full adoption), the expected contrail warming reduction due to contrail avoidance is likely to be 15 times the expected $CO_2$ warming penalty in 2050, and 3 times the expected $CO_2$ warming penalty in 2100. In the more likely scenario of 0.35% fuel burn penalty, the expected contrail warming reduction is >425 times the expected $CO_2$ warming in 2050, and 86 times the expected $CO_2$ warming in 2100.

Over any practical timescale (i.e. within this century), the expected warming due to the fuel burn penalty of contrail avoidance is two orders of magnitude lower than the expected contrail warming reduction. Even in a scenario that assumes an extremely pessimistic fuel burn penalty of 10% (the current literature suggests that a fuel burn penalty of <2% is expected[10]), the probability of a net contrail warming reduction in 2100 is >90%.

## Summary

Reduced order climate modelling is used to assess the impact of contrail avoidance on global surface temperature change towards 2050. We define "effectiveness" as the reduction in global contrail effective radiative forcing (ERF) due to a fleet-wide mitigation measure. Various contrail avoidance scenarios are investigated, including scenarios with different contrail effective radiative forcing (ERF) growth rates beyond 2045, levels of effectiveness of 0%, 25%, 50%, 75%, and 100%, dates of first action of 2035, 2040 and 2045, and levels of fleet averaged fuel burn penalties of 0.35%, 5% and 10% at full adoption. The contrail warming reduction impacts of contrail avoidance and modified fuel composition are also compared, and the combined impacts are explored.

To summarise the main findings of this work, the 2050 temperature impact of a 100% effective contrail avoidance scenario with a 2035 start date is presented in Fig. 7. Even with a pessimistic fuel burn penalty of 10%, the mean temperature reduction of contrail avoidance is 63 times the increase due to fuel burn. The combined temperature

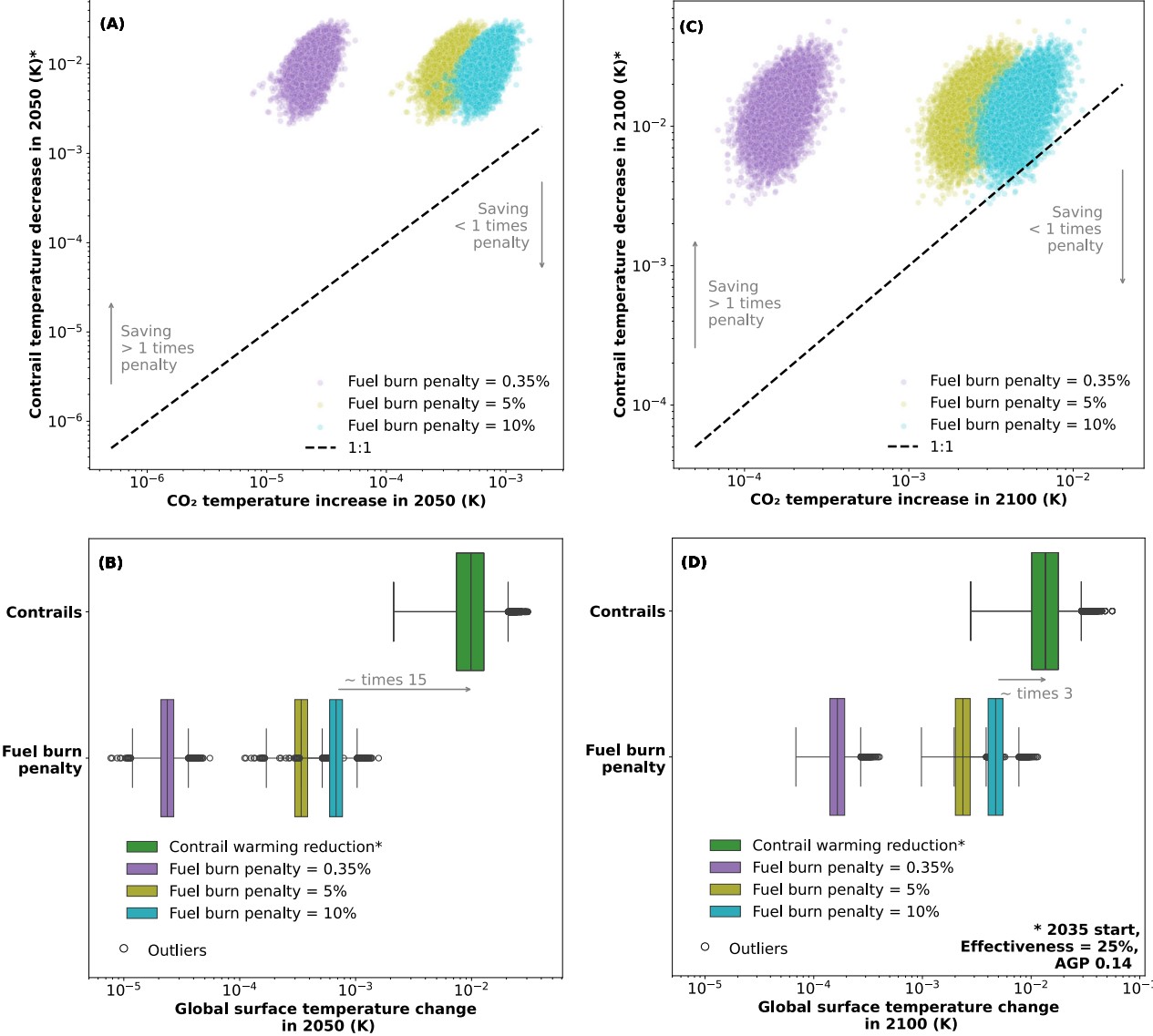

**Fig. 6 | Distribution of the decrease in global surface temperature rise due to contrail avoidance and the temperature increase due to additional fuel burn in 2050 and 2100.** The distributions are shown for: 2050, (**A**) in a scatter plot and (**B**) in a box plot, and 2100, also (**C**) in a scatter plot and (**D**) in a box plot. Contrail avoidance is assumed 25% effective, the scenario start date is assumed to be 2035 and three levels of fuel burn penalty are presented: 0.35%, 5% and 10% of fleet averaged fuel burn at full adoption. Lines labelled "1:1" are shown in the scatter plots, which represent the level where the reduction in contrail warming due to contrail avoidance is equal to the $CO_2$ warming incurred due to the fuel burn penalty. The error bars, shaded areas, and central vertical line in the box plots indicate the 95% confidence intervals, the interquartile range, and the mean, respectively. The aviation growth projection (AGP) AGP 0.14 is assumed, i.e. no growth in flown distance beyond 2045 and a saturation in future contrail forcing effects.

reduction due to contrail avoidance and the contrail-related impacts of modified fuel composition is a mean of 1.6% more than that of contrail avoidance in isolation. Regardless of the fuel composition utilised throughout the fleet therefore, and even if a high fuel burn penalty is permitted, contrail avoidance provides an opportunity to eliminate a substantial proportion of contrail warming by 2050, here equivalent to 9% of the remaining global temperature budget outlined by the Paris Agreement[33].

In this investigation, the same ERF impact from contrails and $CO_2$ is assumed to lead to the same rise in global surface temperature. If instead, the values in Bickel et al.[25] are utilised, 1 W m⁻² of ERF due to contrails has the same impact on global surface temperature as approximately 0.4 W m⁻² of ERF from $CO_2$. The global surface temperature impacts of contrails are then approximately 40% of the results presented here, however the conclusions drawn from this analysis

remain unchanged. In other words, the only contrail avoidance scenarios that have a have a significant (>2%) probability of producing a net warming climate impact within this century have pessimistic (>5%) fleet-averaged fuel burn penalties, low (<25%) levels of effectiveness, and occur in cases where the rate of aviation demand growth beyond 2045 is less than linear. In all other cases, the mean contrail warming reduction due to contrail avoidance is approximately two orders of magnitude higher than the warming penalty due to additional fuel burn.

Centrally, this work highlights the opportunity loss due to delays in contrail avoidance action. A 10-year delay in implementation diminishes the mean temperature reduction detailed in Figure 7 to 2% of the remaining temperature budget, approximately equivalent to a 78% loss in contrail avoidance effectiveness. In short, this analysis identifies inaction as the most significant climate risk associated with contrail avoidance.

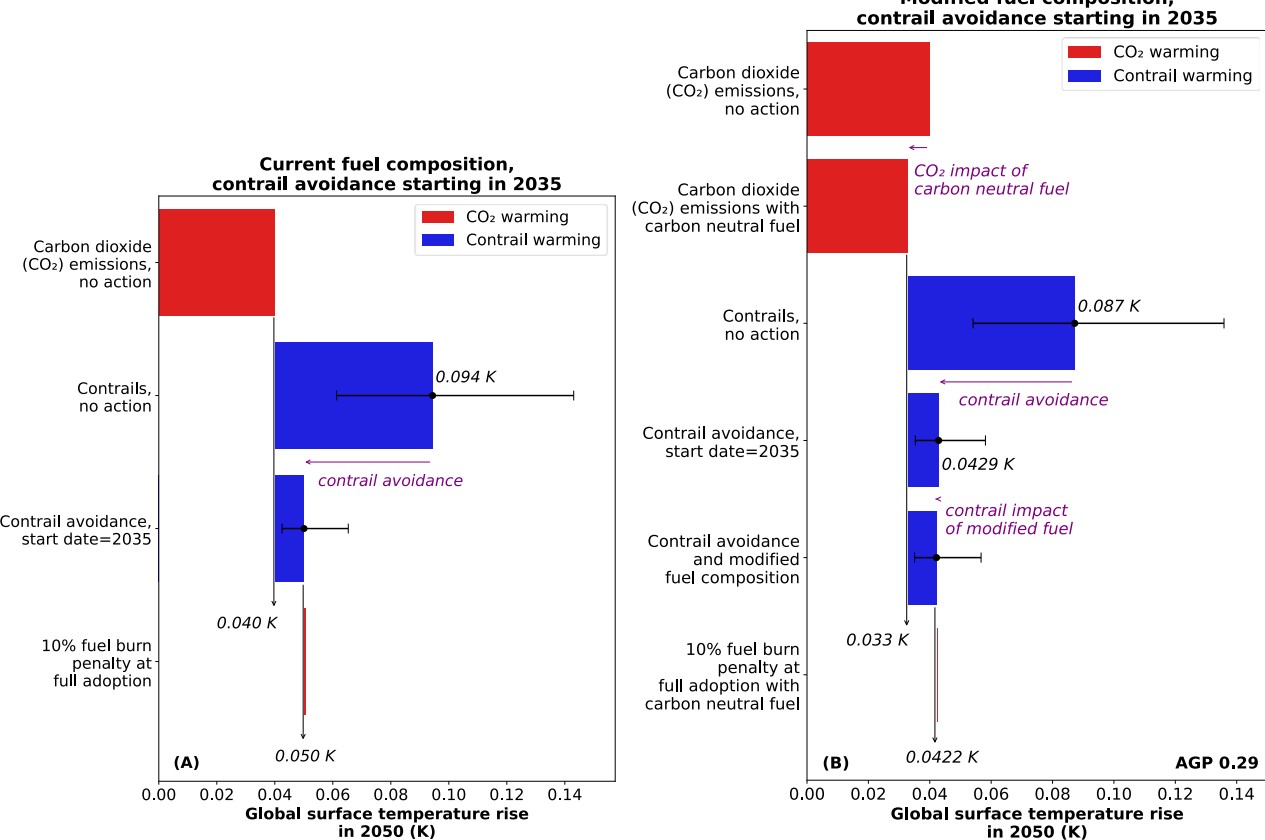

**Fig. 7 | A summary of global surface temperature change in 2050, due to contrail avoidance and its associated fuel burn penalty.** The scenario is presented (**A**) without and (**B**) with modifications to fuel composition. A start date of 2035, a 10-year period between start date and full adoption, a 100% contrail avoidance effectiveness, and a 10% fuel burn penalty at full adoption are assumed. The temperature impact of all aviation carbon dioxide emissions is reduced (but not eliminated) by the introduction of carbon neutral fuel, which is assumed to be phased in alongside the modifications to fuel composition, starting in 2035. All error bars indicate 95% confidence intervals. The aviation growth projection (AGP) AGP 0.29 is assumed, i.e. exponential growth in flown distance beyond 2045 and a saturation in future contrail forcing effects.

## Methods

The climate model used in this analysis requires estimates of the $CO_2$ emissions that can be attributed to aviation, the fleet-wide kilometres flown annually, and the relationship between the ERF due to contrail cirrus and the fleet-wide kilometres flown annually. These values are estimated historically from 1940 (the first year of significant aviation activity) to 2018, and projected into the future up to the year 2100 for various different aviation growth projections or AGPs (see the Aviation growth projections section), assuming no contrail avoidance-based action or modifications to fuel composition. The emission and contrail ERF profiles are adjusted to account for the contrail and $CO_2$ emission ERF changes to contrail avoidance and fuel composition modifications (see the Contrail avoidance and the Changes to fuel composition sections). The climate model uses the scenarios that involve both no mitigation action and mitigation action to estimate annual contrail and $CO_2$ effective radiative forcing (ERF) and global surface temperature responses (see Aviation climate Modelling). We obtain an estimate of the temperature changes due to contrail avoidance and modifications to fuel composition by finding the difference between the temperature responses given no mitigation action and mitigation action.

### Aviation growth projections

The historical annual $CO_2$ emissions of the aviation fleet are estimated using the same methods in Lee et al.[3]. The tank-to-wing $CO_2$ emissions are obtained from Sausen and Schumann[44] for 1940 to 1970, the International Energy Agency[45] for 1971 to 1989, and directly from the Supplementary Materials in Lee et al.[3] for 1990 to 2018. To obtain the well-to-wing $CO_2$ emissions, it is assumed that the total well-to-wing $CO_2$ intensity of Jet-A is 89 kg $CO_2$ (kg fuel)$^{-1}$, and that 82.3% of the $CO_2$ is emitted during the flight[46]. Therefore, the tank-to-wing $CO_2$ emissions are multiplied by 1.215 to obtain annual well-to-wing $CO_2$ emissions from aviation.

The profiles of annual $CO_2$ emissions and kilometres flown from 2018 to 2045, inclusive of the demand depression surrounding the COVID-19 pandemic, are based on scenarios generated by the International Council on Clean Transportation[32]. The annual kilometres flown are assumed to increase by 3% per year in 2019, and from 2024 to 2045. Due to improvements in flight efficiency over time, the annual $CO_2$ emissions do not increase at the same rate as annual kilometres flown. They are assumed to increase by 1.47% per year in 2019, and from 2024 to 2034. The annual $CO_2$ emissions increase by 0.72% per year from 2034 to 2045.

Current literature sources [32,47,48] estimate the growth of aviation demand up to, but not beyond, 2045. We here consider projections of flown distance beyond 2045. Firstly, the same growth rate as up to 2045 is assumed, meaning that the annual kilometres flown and $CO_2$ emissions increase exponentially at a rate of 3% and 0.72% per year beyond 2045, as in AGP 0.36 and AGP 0.29. Secondly, a linear increase in kilometres flown at the same rate as the annual increase from 2044 to 2045 is considered, as in AGP 0.21. In this scenario, the same rate of improvement in annual efficiency is assumed, so that the growth in annual $CO_2$ emissions decreases by 2.28% per year. In other words, the

annual CO2 emissions increased by 97.72% from the previous year. Thirdly, no growth in annual kilometres flown or efficiency improvements are assumed, as in AGP 0.14.

Similar to the annual $CO_2$ emissions profile, the historical ERF contrail trend is obtained directly from Lee et al.[3] for 1990 to 2018 and is assumed to linearly increase from zero for 1940 and 1989. In the scenario where air traffic increases 4-fold from 2006 to 2050, Bock and Burkhardt 2019[27] estimate that the ERF due to contrail cirrus increases from 49 mW m$^{-2}$ to (at least) 159 mW m$^{-2}$. As a conservative scenario therefore, the future ERF of contrails is assumed to scale with the 0.847$^{th}$ power of annual kilometres flown, which corresponds with the aforementioned trend in Bock and Burkhardt 2019[27]. This scenario is denoted by the term "saturation", as the contrail ERF "saturates" rather than linearly increasing with kilometres flown (as in AGP 0.29, AGP 0.21 and AGP 0.14). Since several existing works[2,3,24] assume instead that the future ERF of contrails increases linearly with the annual kilometres flown, a linear increase is also explored, denoted by the phrase "no saturation" (as in AGP 0.36).

To summarise, four distinct aviation growth projections (AGP 0.36, AGP 0.29, AGP 0.21 and AGP 0.14) are explored in this study, which each represent different growth rates in annual kilometres flown and $CO_2$ emissions beyond 2045, and different relationships between annual kilometres flown and contrail ERF. While all AGPs are presented in the Supplementary information, The impact of different contrail avoidance scenarios section explores the opportunity of contrail avoidance using AGP 0.29, and The risk associated with contrail avoidance section explores risk by considering AGP 0.14.

## Contrail avoidance

Contrail avoidance involves small deviations of individual flight paths around contrail forming regions, which reduces the length of persistent contrails that are formed and in turn reduces the annual contrail ERF. In addition, the additional fuel that is consumed due to the small deviations increases the annual $CO_2$ emissions of aviation. In the scenarios where contrail avoidance is implemented, but no changes to fuel composition are assumed, the annual contrail ERF ($ERF_{current\,fuel}$) can be expressed as in Eq. (1):

$$ERF_{current\,fuel}(t) = ERF_{no\,action}(t) \cdot (1 - R(t)) \tag{1}$$

In the contrail ERF profile, the year (i.e. any year between 1940 and 2100) is represented by $t$. The term $ERF_{no\,action}$ represents the annual ERF due to contrail cirrus in the "no action" scenario. The term $R$ is the profile of annual contrail ERF reduction, expressed as a percentage of the annual ERF due to contrail cirrus in the "no action" scenario. A $R$ value of 0% represents the scenario where no contrail avoidance is implemented, and a $R$ value of 100% represents a fully implemented, fully effective contrail avoidance strategy.

The continuous scale-up of contrail avoidance, from first action to fleet wide contrail avoidance, is modelled using an s-curve, as in Eq. (2):

$$R(t) = \begin{cases} t, & 0 < t_0 \\ \eta \cdot (1/(1 + \exp(-1/\Delta t \cdot (100 - \Delta t)/10 \cdot (t - (t_0 + \Delta t/2)))), & t \geq t_0 \end{cases} \tag{2}$$

The scale-up period of the s-curve is represented by the term $\Delta t$. In all scenarios analysed in this study, scale-up is assumed to occur over 10 years. The term $t_0$ represents the first year of contrail avoidance action. The three scenarios analysed in this study have years of first action of 2035, 2040, and 2045, respectively. The term $\eta$ represents the effectiveness of contrail avoidance at full implementation (i.e. the reduction in annual global contrail ERF impact). If $\eta = 100\%$, all of the annual global contrail ERF impact is avoided at full implementation, and if $\eta = 0\%$, none of the impact is avoided. Effectiveness values of 0%, 25%, 50%, 75% and 100% are used in this analysis.

The annual well-to-wing $CO_2$ emissions profile in the scenarios where contrail avoidance is implemented but carbon neutral fuel is utilised ($\dot{m}_{current\,fuel}$), is expressed as in Eq. (3):

$$\dot{m}_{current\,fuel}(t) = \dot{m}_{no\,action}(t) \cdot (1 + FBP(t)) \tag{3}$$

The profile $\dot{m}_{no\,action}$ represents the well-to-wing $CO_2$ emissions in the scenario where no action is taken to avoid contrails, and $FBP$ represents the increase in fuel burn penalty per year due to contrail avoidance, as a percentage of the total annual fuel burn. It increases in value at the same rate as reduction in persistent contrails formed, as in Eq. (4):

$$FBP(t) = \begin{cases} t, & 0 < t_0 \\ FBP_{full} \cdot (1/(1 + \exp(-1/\Delta t \cdot (100 - \Delta t)/10 \cdot (t - (t_0 + \Delta t/2)))), & t \geq t_0 \end{cases} \tag{4}$$

The fuel burn penalty at full scale up is represented by $FBP_{full}$. In this study, $FBP_{full}$ values of 0.35%, 5% and 10% are assumed.

## Changes to fuel composition

Modifications to fuel composition may reduce aviation $CO_2$ emissions but may also affect the formation and effective radiative forcing of persistent contrails. Here these effects are combined by assuming the introduction of a carbon neutral fuel that reduces persistent contrail formation (referred to here as "modified fuel"). The latter effect is represented as a change in contrail ERF, without ($ERF_{\Delta fuel\,only}$) and combined with ($ERF_{\Delta fuel}$) the effects of contrail avoidance, as in Eqs. (5) and (6). The impact of the fuel is considered independent of the impact of contrail avoidance, which is to say that the effects of contrail avoidance and carbon neutral modified fuel are here combined by multiplying the fractional changes in contrail ERF and fuel consumption for each:

$$ERF_{\Delta fuel\,only}(t) = ERF_{no\,action}(t) \cdot \left(1 - F_{\Delta fuel}(t) + M_{\Delta fuel} \cdot F_{\Delta fuel}(t)\right) \tag{5}$$

$$ERF_{\Delta fuel}(t) = ERF_{current\,fuel}(t) \cdot \left(1 - F_{\Delta fuel}(t) + M_{\Delta fuel} \cdot F_{\Delta fuel}(t)\right) \tag{6}$$

The term $F_{\Delta fuel}$ represents the fraction of the annual aviation fuel consumption that can be attributed to the carbon neutral, modified fuel. The $F_{\Delta fuel}$ profile is derived from the Biomass and Power-to-Liquid scenario in Dray et al.[24], meaning that $F_{\Delta fuel} = 0\%$ in 2024, $F_{\Delta fuel} = 6\%$ in 2030, $F_{\Delta fuel} = 50\%$ in 2040 and $F_{\Delta fuel} = 100\%$ in 2050 and the years after. The term $M_{\Delta fuel}$ represents the total contrail effective radiative forcing in a scenario where only the modified fuel is utilised throughout the fleet, relative to one where the current fuel composition is utilised. It is also approximated from Dray et al.[24] as 58%.

As previously mentioned, the modified fuel is also assumed to be carbon neutral, which means that the net annual well-to-wing $CO_2$ emission of a fleet that only utilises the modified fuel is assumed to be zero. The annual well-to-wing $CO_2$ emission of a fleet where the fuel composition is modified over time is therefore expressed without ($\dot{m}_{\Delta fuel\,only}$) combined with ($\dot{m}_{\Delta fuel}$) the effects of contrail avoidance, as in Eqs. (7) and (8):

$$\dot{m}_{\Delta fuel\,only}(t) = \dot{m}_{no\,action}(t) \cdot \left(1 - F_{\Delta fuel}(t)\right) \tag{7}$$

$$\dot{m}_{\Delta fuel}(t) = \dot{m}_{current\,fuel}(t) \cdot \left(1 - F_{\Delta fuel}(t)\right) \tag{8}$$

## Aviation climate modelling

Aviation's contribution to climate change, quantified through Effective Radiative Forcing (ERF) and subsequent change in global surface

temperature, is modelled using the Aviation Climate and Air quality Impacts (ACAI) model. This model is a Julia language implementation of the Aviation environmental Portfolio Management Tool - Impacts Climate (APMT-IC)[2,32]. ACAI computes these climate impacts for uncertain input parameters using a quasi-Monte Carlo method with 10,000 members, as detailed in Grobler et al.[2].

To align the model with the recent scientific consensus, three model updates are implemented. Firstly, we align the $CO_2$ impulse response functions and background $CO_2$ concentrations to match most recent SSP scenarios obtained from the IPCC 6th Coupled Model Intercomparison Project (CMIP6), described by the Intergovernmental Panel on Climate Change (IPCC) Sixth Assessment Report (AR6). In this study, we model $CO_2$ based on a background SSP scenario of SSP1-2.6, representing a low background $CO_2$ emissions future scenario. Using this SSP scenario, impulse response functions for each year, and background $CO_2$ concentrations are derived using the Model for Greenhouse Gas Induced Climate Change version 6 (MAGICC6)[2,49].

Secondly, the ACAI temperature model is updated to align with the IPCC Sixth Assessment Report's (AR6) findings, based on the Coupled Model Intercomparison Project Phase 6 (CMIP6). Consistent with the Finite Amplitude Impulse Response (FAIR) v2.0 framework temperature model[50], we implement a three-timescale impulse-response function. This approach applies three distinct exponential decay terms to represent the Earth system's thermal response. The model's prior uncertainty distribution parameters are chosen to represent the range of thermal response characteristics observed across the CMIP6 model outputs. Subsequently, parameter sub-sampling is performed to constrain the model's output against historical temperature observations. This three-timescale approach improves the emulation of the climate complexity observed in the CMIP6 results. This method, including parameter selection and calibration, is detailed in Leach et al.[50]. It yields an Equilibrium Climate Sensitivity (ECS) of 3.66 K (mean) and 5th to 95th percentile range of 1.94 K to 6.59 K. It also yields a Transient Climate Response (TCR) of 1.82 K (mean) and 5th to 95th percentile range of 1.30 K to 2.45 K.

The effective radiative forcing of contrails was updated to align more closely with the uncertainty distribution in Lee et al.[3]. Contrail impacts are linearly scaled to RF values from Lee et al.[3]. The contrail ERF/RF distribution is based on values from Lee et al.[3]. However, unlike Lee et al.[3], this study separates the uncertainty from RF and the uncertainty in ERF/RF. The RF uncertainty distribution is directly from Lee et al.[3] and is taken as a triangular distribution with minimum, midpoint, and maximum values of $0.674 \times 10^{-9}$, $2.25 \times 10^{-9}$, and $3.82 \times 10^{-9}$ mW m$^{-2}$ (flight-km)$^{-1}$, respectively. The ERF/RF uncertainty distribution is also taken as a triangular distribution, with minimum, midpoint, and maximum values of 0.31, 0.351, and 0.59 (mW m$^{-2}$)/(mW m$^{-2}$), respectively. This results in an average ERF/RF of 0.42 (mW m$^{-2}$)/(mW m$^{-2}$), matching the average applied by Lee et al.[3]. Collectively, for $52.61 \times 10^9$ flight-km in 2018[3], this results in an ERF of 57.4 mW m$^{-2}$ (2.5% to 97.5% percentile range of 25.0 mW m$^{-2}$ to 97.4 mW m$^{-2}$), which is similar to the ERF range reported by Lee et al.[3] of 57.4 mW m$^{-2}$ (range 17.5 mW m$^{-2}$ to 97.6 mW m$^{-2}$). Differences in the uncertainty ranges can be explained by the explicit separation of RF and ERF/RF uncertainties and the use of a quasi-Monte Carlo simulation to derive the subsequent ERF of contrails.

## Data availability

The input aviation data associated with this work has been deposited under accession code https://doi.org/10.5281/zenodo.18245429. The output climate data associated with this work have been deposited under accession codes https://doi.org/10.5281/zenodo.18246901, https://doi.org/10.5281/zenodo.18256701, https://doi.org/10.5281/zenodo.18256763, and https://doi.org/10.5281/zenodo.18256815. Extended versions of the figures in this study are provided in the Supplementary information.

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

## Acknowledgements

While entirely different in content, the authors of this work collaborated closely with the authors of the Aviation Impact Accelerator's online report, "The 2030 Sustainable Aviation Goals: Five Years to Chart a New Future for Aviation", which is available at https://report.aiazero.org/. We would therefore like to thank Robert Miller, Eliot Whittington, Samuel Gabra, Jay Green, Jia Wei Kho and Deepanshu Singh for their continued advice and support.

## Author contributions

This paper was conceptualised by S.R.H.B. S.R.H.B., M.E.J.S., M.L.S., S.D.E., P.J.H., J.M., and M.M. directed the narrative and scope of the paper. The models that feature in this work were derived by J.R.S., P.J.H., M.E.J.S., J.M., C.G., and S.D.E. Climate modelling was completed by C.G., and data visualisation and output analysis were completed by J.R.S. The initial draft of this work was completed by J.R.S. Reviewing and editing was completed by J.R.S., C.G., P.J.H., J.M., M.L.S., M.M., M.E.J.S., S.D.E., and S.R.H.B.

## Competing interests

The authors declare no competing interests.
