## [Transparent Peer Review file · Nature Communications]

The Climate Opportunities and Risks of Contrail Avoidance

Corresponding Author: Dr Jessie Smith

Version 0:

Reviewer comments:

Reviewer #1

(Remarks to the Author)

Summary

=====

The authors present a study on the opportunities and risks of contrail avoidance and compare it with the use of sustainable aviation fuels.

The paper is well-written and the approach relies on simplifications of various atmospheric processes to allow for a scenario assessment and to cope with uncertainties, monte-carlo simulations are performed.

As it provides a guidance for a aviation stakeholder it is very important. However, there are some major issues with the method used.

1) eq. (1) would indicate that every flown kilometre leads to a contrail. That is obviously wrong and I suppose that the description is misleading at best.

2) the assumptions made in the simulation set-up are towards a larger contrail effect.

Without clarifying these aspects I would not support a publication.

Major points (summary, more details given below):

=====

1) Some assumptions are biased towards a larger contrail effect:

- No saturation effects in contrails coverage is considered (see also below the BB19 discussion)
- The aviation scenario has an unlimited growth. Therefore the short-term effects are getting more important that in an aviation scenario that e.g. stabilises at a certain value.
- The effectiveness of the contrail avoidance ranges between 70% to 100%, while other authors claim that it also could be negative for many individual cases and hence the overall effectiveness might be much lower.
- The contrail avoidance reduction percentage is calculated based on a no-SAF case, which makes it larger than having first a contrail impact reduction due to SAF and then calculating the avoidance reduction.
- The efficacy was discussed, but taking that together with the four above mentioned points what is the impact?

2) Although the method looks good in principle, I think there is a flaw in equation 1, as it states that every flown kilometre leads to a contrail.

3) Lack of literature leads to partial exaggerations

"Critically, no

existing literature is known to contextualise the climate impacts of navigational contrail avoidance in terms of global surface temperature rise."

I think this is just not true (see below).

Line-by line remarks

=====

Page 3:

"Critically, no existing literature is known to contextualise the climate impacts of navigational contrail avoidance in terms of global surface temperature rise."

I am not quite sure if this is fully correct. Surface temperature rise is a common metric for a couple of papers. Enclosed are 4 papers, but I think there are many more.

Fig. 5 of Schumann et al. 2011, Potential to reduce the climate impact of aviation by flight level changes, 3rd AIAA Atmospheric Space Environments Conference 27 - 30 June 2011, Honolulu, Hawaii, AIAA 2011-3376.

Grewe et al., 2017, ERL write:

"We use the global and temporal average near-surface temperature response over 20 years after introducing the climate-optimized routing strategy."

Meuser et al. MITIGATION OF AVIATION'S CLIMATE IMPACT THROUGH ROBUST CLIMATE OPTIMIZED TRAJECTORIES IN INTRA-EUROPEAN AIRSPACE (2022) 33rd Congress of the International Council of the Aeronautical Sciences, ICAS 2022, 9, pp. 6553 - 6567,

Simorgh et al. 2023 Robust 4D climate-optimal flight planning in structured airspace using parallelized simulation on GPUs: ROOST V1.0, <https://doi.org/10.5194/gmd-16-3723-2023>

page 4

"The contrail temperature rise distribution"

I would appreciate a more precise language. Not the temperature of the contrail is changes but the air temperature near the surface due to contrails

page 5 / page 13

"This

is because contrail warming is assumed to scale with annual kilometres flown."

"Bock and Burkhardt 2019 estimate the ERF of contrails into the future. While their findings are uncertain, the mean ERF of contrails scales approximately linearly with aviation demand growth"

Why is this an appropriate assumption. The contrail coverage is limited by the potential contrail coverage. And saturation effects are significant.

Bock and Burkhardt 2019 show that for an increase in air traffic from 2006 to 2050 by a factor of 4.3 the increase in RF by contrails is only 3.26. This is a saturation by more than 20%.

page

"Hypothetically, if all contrail warming was eliminated by 2050, this would be equivalent to a temperature saving of 11.6% [1.8%, 13.0%] of the remaining global temperature budget. This alone demonstrates the opportunity for rapid warming reduction that contrail avoidance presents."

The calculation is unclear to me.

- I think the rate of elimination is important to the temperature impact.

"rapid warming reduction": The warming reduction is not instantaneously, but includes the atmospheric-ocean inertia. So please elaborate a bit more on the time scales that are involved.

page 8

"The single greatest contrail warming reduction therefore occurs when contrail avoidance is implemented in 2035. Modified fuel composition, and the improvement of contrail avoidance effectiveness also have an impact. If the primary goal is contrail warming mitigation however, such changes should not be prioritised over early contrail avoidance action."

page 7/9

"contrail avoidance effectiveness (70%, 85% and 100%)" and "0.35% is the closest to the distribution of values evidenced in

recent literature^{9,31,32}, and 5% and 10% fuel burn penalty represent two different levels of pessimism for heuristic purposes^{33–39}."

The assumption in most of the studies, if not all, is a well-known atmosphere, which is acceptable from a modelling perspective

to obtain a theoretical limit for the mitigation potential. They are not quantifying the uncertainties arising from weather forecasts or the ability to forecast ISSRs.

However, here in this paper the consequences for the real world application are discussed and hence such uncertainties should be taken into account.

This is, e.g. discussed in Lee et al. 2023 [16] and Grewe et al. 2017 [37]. What if the contrail effectiveness is only 40%, because

a) the weather forecast is incorrect and a rerouted flight actually goes through a contrail region instead of avoiding it?

b) the contrail impact of the avoided contrail was estimated positive, but actually a cooling contrail was avoided,

c) ATC limitation inhibit the avoidance of the contrail region.

page 14

"3% per year in 2019, and from 2024 onwards. "

This is an exponential growth in aviation. This does not look realistic, right?

The assumption of an unlimited growth has an impact on the importance of non-CO₂ to CO₂ impacts and put a increases the importance of short-term effects, i.e. non-CO₂, contrail effects.

Hence, the results are biased towards contrails effects due to the assumption of the unlimited growth.

I think a sensitivity with a non-exponential (linear) and a scenario that is e.g. constant after e.g.

2040 would be necessary to avoid this bias.

page 14

Formulae 1

If no contrail avoidance is implemented ($R=0$) then the persistent contrail length ($L_{\text{current fuel}}$) is equal to the flown kilometres (L_{flight}).

This is obviously not the case.

page 16

"reduced contrail fuel."

Not quite sure what this means.

page 16

eq 5:

I think, the equation is biased: $M_{\text{Delta-fuel}}$ is the percentage reduction with the reference for current fuel.

Hence, it is based of a difference where you first identify the reduction potential in contrail avoidance, before considering other effects.

By this approach the percentages are the largest. If one would use SAF for targeted contrail regions first and then consider contrail avoidance,

the results would be the opposite of your findings.

page 17

the ECS is presented as unitless. Shouldn't it be in K?

References incomplete:

IEA. IEA, Oil Information. (2024).

Klima, K. Assessment of a Global Contrail Modeling Method and Operational Strategies for Contrail Mitigation.

Reviewer #2

(Remarks to the Author)

Thank you for the opportunity to review the paper "The Climate Opportunities and Risks of Contrail Avoidance,"

The paper evaluate the effect on global temperature change based on operationalization of a mitigation strategy: contrail avoidance.

This is a relevant topic. The methodology applied is appropriate. The results are reasonable.

The paper is well written.

Section: The Impact of Different Contrail Avoidance scenarios. The paper describes the impact of Different Start Dates (2035, 2040, 2045), Different Levels of Effectiveness, Modification of Fuel Composition

1) page 5-6, What is the exact definition of "Effectiveness"? What factors impact effectiveness? Does the ability to accurately identify ISS regions affect effectiveness?

2) Use of Matrices to visualize results. For Figures 2,3,4 would a matrix with cells of 2055 Temp Rise and columns and rows with the Factors make the results easier to summarize ? For example Figure 2 and 3: Row start years and Columns Effectiveness. Figure 4: Rows start year, columns Current Fuel, Modified Fuel.

3) Be clear which factor, Start Date, Effectiveness, or Fuel has the biggest influence?

If you had to pick one strategy, which would it be? Why?

More realistically, where would you recommend the research funding be spent: Start Date, Effectiveness, or Fuel?

4) Recommend explain why the Start Date is so important? It integrates over time!

5) Recommend a short discussion on the issue of accurately identifying the geographic and vertical location of Ice Super Saturated regions. See #1 above. What is current accuracy (e.g. from flight trials? What is current state-of-the-art in detection? How could this be improved?

What about forecasting?

6) Methods Section: OK.

Version 1:

Reviewer comments:

Reviewer #1

(Remarks to the Author)

Review of the revised version:

The authors have thoroughly revised their manuscript and adequately answered my questions.

Thank you for efforts.

I have some minor, though important, comments that I would like the authors to consider:

I 27-28: The sentence gives the impression that the climate impact of aviation solely arises from CO₂ and contrails. Lee et al. 2021 gives a different view.

I think it would be good to adapt this sentence, accordingly.

I 42 You are referring to a temperature rise and give Lee et al. as a reference(3). However, I think that paper concentrates on RF and ERF rather than dT, correct?

I 402 "Climate Modelling" in order to avoid confusion with GCM modelling I suggest to adapt to something like "simplified climate modelling".

Conclusion: As in the lines 64-68 the term "effectiveness" is also interpreted in terms of forecast abilities, it would be good to explicitly state again in the conclusion

to what this term actually refers. To my understanding, the aspect of forecast abilities or worsening the situation is not covered.

Reviewer #2

(Remarks to the Author)

The Authors have responded to the Review comments satisfactorily.

The paper is improved and easier to read.

This reviewer recommends publication of the paper as-is.

The Climate Opportunities and Risks of Contrail Avoidance – Response to Reviewers

We would like to thank the reviewers for their invaluable feedback on our manuscript. This document provides a detailed response to the provided feedback. It summarises the changes we have made to improve the manuscript and includes direct references to the manuscript text. Some guiding notes for interpreting this document are detailed below:

- *The reviewer comments are detailed in **bold font**. Our point-by-point responses are detailed in non-bold, italicised font.*
- *Any direct quotes from the manuscript are in black, non-italicised text.*
- *Any inserted text is red, non-italicised and underlined.*
- *Any deleted text is not shown.*

Response to Reviewer 1

The authors present a study on the opportunities and risks of contrail avoidance and compare it with the use of sustainable aviation fuels.

The paper is well-written and the approach relies on simplifications of various atmospheric processes to allow for a scenario assessment and to cope with uncertainties, monte-carlo simulations are performed.

As it provides a guidance for a aviation stakeholder it is very important.

However, there are some major issues with the method used.

1) eq. (1) would indicate that every flown kilometre leads to a contrail. That is obviously wrong and I suppose that the description is misleading at best.

2) the assumptions made in the simulation set-up are towards a larger contrail effect.

Without clarifying these aspects I would not support a publication

We would like to thank Reviewer 1 for their detailed and constructive feedback. We have made efforts to address each comment in point-by-point responses. In particular we have:

- Corrected equation 1 to:

$$ERF_{current\ fuel}(t) = ERF_{no\ action}(t) \cdot (1 - R(t)) \quad (1)$$

The mistake in this equation was a transcription issue, which is to say that the incorrect equation was not used in calculation of the results, and the error has no consequences for our results.

- Modified the scenarios that were analysed to explore different contrail ERF and demand growth scenarios into the future. These scenarios are discussed in the additional “Aviation Growth Projections Towards 2100” of the main text, and the “Extended Data” section details additional results that relate to these scenarios.
- Expanded the analysis to incorporate more levels of effectiveness and modified fuel composition settings. Additional discussion has been added to the “The Impact of Different Contrail Avoidance Scenarios” section to discuss these new scenarios.

Major points (summary, more details given below):

1. Some assumptions are biased towards a larger contrail effect:

- **No saturation effects in contrails coverage is considered (see also below the BB19 discussion)**
- **The aviation scenario has an unlimited growth. Therefore the short-term effects are getting more important that in an aviation scenario that e.g. stabilises at a certain value.**

To address these points, we have introduced four distinct aviation growth projections that represent different levels of contrail ERF saturation and demand growth beyond 2045. Two growth projections, AGP 0.36 and AGP 0.29, assume a 3% growth in flown kilometres per year. The former assumes no saturation effects, and the latter assumes saturation effects that align with the contrail ERF growth in Bock and Burkhardt 2019. The AGP 0.21 and AGP 0.14 projections assume linear and no growth in flown kilometres per year, respectively. These two scenarios also

assume saturation effects that align with the contrail ERF growth in Bock and Burkhardt 2019.

The global surface temperature rise towards 2100 that results from these four growth profiles is discussed in the “Aviation Growth Projections Towards 2100” section, which is a new addition to the manuscript. The assumptions behind the four growth projections have been discussed in paragraphs 3-5 of the “Aviation Growth Projections” section of the “Methods”, which are also new additions to the manuscript.

“The Impact of Different Contrail Avoidance Scenarios” section explores the opportunity of contrail avoidance using AGP 0.36, and “The Risk Associated with Contrail Avoidance” section explores risk by considering AGP 0.14. However, each figure in the main text has been generated for all four growth projections, as seen in the “Extended Data” section.

- **The effectiveness of the contrail avoidance ranges between 70% to 100%, while other authors claim that it also could be negative for many individual cases and hence the overall effectiveness might be much lower.**

To fully explore the impact of effectiveness, we now consider 5 levels of effectiveness at 0% (equivalent to the “No Action” scenario), 25%, 50%, 75% and 100%. The impact of the different levels of effectiveness on global surface temperature rise are presented in Figure 3. The impact of effectiveness is referred to throughout the “The Impact of Different Contrail Avoidance Scenarios” section and is discussed in paragraph 2 of this section.

- **The contrail avoidance reduction percentage is calculated based on a no-SAF case, which makes it larger than having first a contrail impact reduction due to SAF and then calculating the avoidance reduction.**

Figure 5 has been modified and paragraphs 7-9 of “The Impact of Different Contrail Avoidance Scenarios” section have been added to explore the impact of contrail avoidance both with and without modified fuel composition.

Equations 5, 6, 7 and 8 now represent the change in contrail ERF and annual well to wing CO₂ emissions of aviation both with and without modified fuel composition.

These equations in combination with equations 1 to 4 now represent contrail ERF and annual well to wing CO₂ emissions, in all combinations of scenarios where contrail avoidance is or is not implemented and the fuel composition is and is not modified.

- **The efficacy was discussed, but taking that together with the four above mentioned points what is the impact?**

We have added some additional sentences to the paragraph 3 of the “Conclusions” section to discuss this. The paragraph now reads:

In this investigation, the same ERF impact from contrails and CO₂ is assumed to lead to the same rise in global surface temperature. If instead, the values in Bickel et al. 2025²⁵ are utilised, 1 W/m² of ERF due to contrails has the same impact on global surface temperature as approximately 0.4 W/m² of ERF from CO₂. The global surface temperature impacts of contrails are then approximately 40% of the results presented here, however the conclusions drawn from this analysis remain unchanged. In other words, the only contrail avoidance scenarios that have a significant (>2%) probability of producing a net warming climate impact within this century have pessimistic (>5%) fleet-averaged fuel burn penalties, low (<25%) levels of effectiveness, and occur in cases where the rate of aviation demand growth beyond 2045 is less than linear. In all other cases, the mean contrail warming reduction due to contrail avoidance is approximately two orders of magnitude higher than the warming penalty due to additional fuel burn.

- 2. Although the method looks good in principle, I think there is a flaw in equation 1, as it states that every flown kilometre leads to a contrail.**

There was indeed a flaw in this equation – we thank the reviewer for identifying it. This equation was not used in calculation of the results, so the error has no consequences for our results. Equation 1, and the text surrounding it (paragraph 1 and 2 of the “Contrail Avoidance” sections of the “Methods”), has been corrected to:

Contrail avoidance involves small deviations of individual flight paths around contrail forming regions, which reduces the length of persistent contrails that are formed and in turn reduces the annual contrail ERF. In addition, the additional fuel that is consumed due to the small deviations increases the annual CO₂ emissions of

aviation. In the scenarios where contrail avoidance is implemented, but no changes to fuel composition are assumed, the annual contrail ERF ($ERF_{current\ fuel}$) can be expressed as in Equation (1).

$$ERF_{current\ fuel}(t) = ERF_{no\ action}(t) \cdot (1 - R(t)) \quad (1)$$

In the contrail ERF profile, the year (i.e. any year between 1940 and 2100) is represented by t . The term $ERF_{no\ action}$ represents the annual ERF due to contrail cirrus in a no action scenario. The term R is the profile of annual contrail ERF reduction, expressed as a percentage of the annual ERF due to contrail cirrus in a no action scenario.

3. Lack of literature leads to partial exaggerations

- "Critically, no existing literature is known to contextualise the climate impacts of navigational contrail avoidance in terms of global surface temperature rise."

I think this is just not true (see below).

This sentence and the paragraphs surrounding it have been modified to avoid partial exaggerations. See the next comment.

Line-by line remarks

Page 3: "Critically, no existing literature is known to contextualise the climate impacts of navigational contrail avoidance in terms of global surface temperature rise."

- I am not quite sure if this is fully correct. Surface temperature rise is a common metric for a couple of papers.
 - Enclosed are 4 papers, but I think there are many more.
 - Fig. 5 of Schumann et al. 2011, Potential to reduce the climate impact of aviation by flight level changes, 3rd AIAA Atmospheric Space Environments Conference 27 - 30 June 2011, Honolulu, Hawaii, AIAA 2011-3376.
 - Grewe et al., 2017, ERL write: "We use the global and temporal average near-surface temperature response over

20 years after introducing the climate-optimized routing strategy."

- **Meuser et al. MITIGATION OF AVIATION'S CLIMATE IMPACT THROUGH ROBUST CLIMATE OPTIMIZED TRAJECTORIES IN INTRA-EUROPEAN AIRSPACE (2022) 33rd Congress of the International Council of the Aeronautical Sciences, ICAS 2022, 9, pp. 6553 - 6567,**
- **Simorgh et al. 2023 Robust 4D climate-optimal flight planning in structured airspace using parallelized simulation on GPUs: ROOST V1.0, <https://doi.org/10.5194/gmd-16-3723-2023>**

The paragraphs 6 and 7 of the introduction (labelled "Main") have been modified. We have added these literature sources detailed above and reworded the final sentence so as to avoid partial exaggerations. The paragraphs now read:

Previous works relevant to contrail warming mitigation primarily focus on either the climate impact of contrails, or the effectiveness of contrail mitigation measures^{24,25}. Various studies explore the climate impact of contrails in terms of their ERF. Lee et al. 2021³ compares recent ERF from historical aviation emissions, including CO₂, contrails, and other non-CO₂ effects, while Bock and Burkhardt 2019²⁶ focuses on estimating the ERF of contrails into the future. Klöwer et al. 2021⁴ instead analyses the global surface temperature change due aviation CO₂ and non-CO₂ under various future aviation emission scenarios. Bickel et al. 2025²⁷ concludes that the ERF impact from contrails likely has a lower global surface temperature impact than the same ERF from CO₂. While the ERF reduction due to technological change and the use alternative fuels are accounted for, contrail avoidance is not included in these works.

Conversely, Dray et al. 2022²³ estimates the ERF reduction of modified fuel composition and a single contrail avoidance strategy with an assumed effectiveness and fuel burn penalty. Grewe et al. 2017¹³, Meuser et al. 2022²⁸ and Simorgh et al. 2023²⁹ analyse the impact of climate optimal routing, which includes contrail avoidance alongside minimising CO₂ and non-CO₂ emissions such as nitrogen oxides. Frias et al. 2024⁹, Teoh et al. 2020⁸, and Smith et al. 2025¹⁸ explore the ERF

reduction and fuel burn penalty associated with various contrail avoidance strategies. Schumann et al. 2011³⁰ investigates the global surface temperature rise due to a contrail avoidance strategy, but remarks that the work to integrate of the presented results into a climate model is ongoing. Critically, no existing literature is known to use climate modelling to analyse the climate impacts of contrail avoidance at various levels of effectiveness, dates of first action, and fuel burn penalties, so as to comprehensively investigate the opportunities and risks that contrail avoidance presents.

page 4: "The contrail temperature rise distribution"

- **I would appreciate a more precise language. Not the temperature of the contrail is changes but the air temperature near the surface due to contrails**

We have removed all instances where the “contrail temperature rise distribution” is referred to and replaced them with “distribution in global surface temperature rise due to contrails”. See, for example, paragraph 1 of the “Aviation Growth Projections Towards 2100” section.

page 5 / page 13 "This is because contrail warming is assumed to scale with annual kilometres flown." "Bock and Burkhardt 2019²⁴ estimate the ERF of contrails into the future. While their findings are uncertain, the mean ERF of contrails scales approximately linearly with aviation demand growth"

- **Why is this an appropriate assumption. The contrail coverage is limited by the potential contrail coverage. And saturation effects are significant.**
- **Bock and Burkhardt 2019 show that for an increase in air traffic from 2006 to 2050 by a factor of 4.3 the increase in RF by contrails is only 3.26. This is a saturation by more than 20%.**

We have analysed further scenarios that address the impacts of saturation according to the values in Bock and Burkhardt 2019. Please see the response to the first of the “Major points” outlined by Reviewer 1.

Page XX "Hypothetically, if all contrail warming was eliminated by 2050, this would be equivalent to a temperature saving of 11.6% [1.8%, 13.0%] of the

remaining global temperature budget. This alone demonstrates the opportunity for rapid warming reduction that contrail avoidance presents."

- **The calculation is unclear to me.**
 - **I think the rate of elimination is important to the temperature impact.**
 - **"rapid warming reduction": The warming reduction is not instantaneously, but includes the atmospheric-ocean inertia. So please elaborate a bit more on the time scales that are involved.**

The term "rapid" is indeed subjective. We have removed all mentions of this and clarified that "atmosphere-ocean inertia" is included in the assumption. Paragraphs 3 and 4 of "The No Action Scenario" section now read:

The 2015 Paris Agreement aims to limit global surface temperature rise to less than 2 K above pre-industrial levels^{25,32}. The current global surface temperature rise above pre-industrial levels is approximately 1.4 K³³. If all anthropogenic emissions were to cease instantly, the IPCC predicts that global surface temperatures would peak at approximately 0.1 K above the temperature at that time (Figure 1.5 of the Global Warming of 1.5 °C Special Report³⁴). The committed temperature rise above industrial levels is therefore estimated to be 1.5 K, which means that the remaining global surface temperature budget is 0.5 K. Beyond this, the 2 K Paris Agreement limit would be exceeded.

Hypothetically, if all contrail warming was eliminated by 2050 (accounting for both the implementation of contrail avoidance and the atmospheric-ocean inertia in global surface temperature rise), this would be a reduction in global surface temperature rise equivalent to 11% of the remaining global temperature budget. This demonstrates the opportunity for warming reduction that contrail avoidance presents.

page 8 "The single greatest contrail warming reduction therefore occurs when contrail avoidance is implemented in 2035. Modified fuel composition, and the improvement of contrail avoidance effectiveness also have an impact. If the primary goal is contrail warming mitigation however, such changes should not be prioritised over early contrail avoidance action."

This paragraph has been revised, since the warming reduction does indeed depend on the level of contrail avoidance effectiveness. Paragraph 10 of “The Impact of Different Contrail Avoidance Scenarios” section now reads:

Therefore, a contrail avoidance scenario which is first implemented in 2035 is expected to achieve a greater contrail warming reduction than modified fuel compositions, provided the effectiveness of contrail avoidance is greater than approximately 32%. This assertion neglects any improvements in effectiveness that may occur as contrail avoidance is adopted throughout the fleet. In such cases, contrail avoidance achieves a greater contrail warming reduction than modified fuel compositions at effectiveness values that are <32%.

page 7/9 "contrail avoidance effectiveness (70%, 85% and 100%)" and "0.35% is the closest to the distribution of values evidenced in recent literature^{9,31,32}, and 5% and 10% fuel burn penalty represent two different levels of pessimism for heuristic purposes^{33–39}."

- **The assumption in most of the studies, if not all, is a well-known atmosphere, which is acceptable from a modelling perspective to obtain a theoretical limit for the mitigation potential. They are not quantifying the uncertainties arising from weather forecasts or the ability to forecast ISSRs.**
- **However, here in this paper the consequences for the real world application are discussed and hence such uncertainties should be taken into account.**
- **This is, e.g. discussed in Lee et al. 2023 [16] and Grewe et al. 2017 [37]. What if the contrail effectiveness is only 40%, because**
 - **a) the weather forecast is incorrect and a rerouted flight actually goes through a contrail region instead of avoiding it?**
 - **b) the contrail impact of the avoided contrail was estimated positive, but actually a cooling contrail was avoided,**
 - **c) ATC limitation inhibit the avoidance of the contrail region.**

Lower levels of effectiveness than originally investigated are indeed possible, so we have revised the manuscript to account for these impacts.

As discussed in the response to the first of the “Major points” outlined by this reviewer, we have conducted some further analysis into impact of effectiveness, detailed in Figure 3 and discussed in detail in paragraph 2 of “The Impact of Different Contrail Avoidance Scenarios” section. We now investigate 5 levels of effectiveness at 0% (equivalent to the “No Action” scenario), 25%, 50%, 75% and 100%.

The exploration of contrail avoidance risk in “The Risk Associated with Contrail Avoidance” section now centres around a lower level of effectiveness than originally investigated: the original submission investigated an effectiveness level of 100% in this section, whereas we now discuss results based on an effectiveness level of 25%.

In addition, we have added some discussion on effectiveness and its limits in paragraph 4 of the introduction. It reads:

We here define “effectiveness” as the reduction in global contrail effective radiative forcing due to a fleet wide mitigation measure. Effective radiative forcing (ERF) is a measure of the climate forcing due to an emission after allowing for tropospheric temperature adjustments. At present, prediction of the geolocation and vertical extent of ISSRs is limited. This in turn imposes a limit on the effectiveness of contrail avoidance since a planned manoeuvre may be a) insufficiently large to avoid the full extent of the ISSR, b) unneeded since an ISSR was incorrectly predicted, or c) perverse since the manoeuvre deviates into an ISSR rather than away from it^{15,16}. There are other examples of factors that can impact the effectiveness of contrail avoidance, such as conflicts between the flight paths of deviating and non-deviating aircraft and limitations on air traffic controller capacity⁸. It is possible to estimate effectiveness of contrail avoidance from flight trial and ISSR forecast data^{10,11,15,17} as in Smith et al. 2025¹⁸. However, there is high uncertainty in this estimate, and there are likely to be technological and operational advancements as further flight trials are conducted. It is therefore prudent to here examine a wide range of effectiveness values to fully explore the climate impacts of contrail avoidance.

page 14 "3% per year in 2019, and from 2024 onwards. " This is an exponential growth in aviation.

- **This does not look realistic, right?**

- The assumption of an unlimited growth has an impact on the importance of non-CO2 to CO2 impacts and put a increases the importance of short-term effects, i.e. non-CO2, contrail effects.
- Hence, the results are biased towards contrails effects due to the assumption of the unlimited growth.
- I think a sensitivity with a non-exponential (linear) and a scenario that is e.g. constant after e.g. 2040 would be necessary to avoid this bias.

We have implemented these suggestions, as explained in more detail in the response to the first point of the “Major Points” detailed by this reviewer.

page 14 Formulae 1 If no contrail avoidance is implemented ($R=0$) then the persistent contrail length ($L_{\text{current fuel}}$) is equal to the flown kilometres (L_{flight}).

- This is obviously not the case.

We have addressed how equation 1 has been corrected in our response to point 2 of the “Major Points” identified by this reviewer.

page 16 "reduced contrail fuel."

- Not quite sure what this means.

In all instances, we have corrected this to “modified fuel” or “modified fuel composition”. See, for example, paragraph 2 of the “Changes to Fuel Composition” section in the “Methods”:

The term $F_{\Delta\text{fuel}}$ represents the fraction of the annual aviation fuel consumption that can be attributed to the carbon neutral, modified fuel.

page 16

- eq 5: I think, the equation is biased: $M_{\Delta\text{fuel}}$ is the percentage reduction with the reference for current fuel.
- Hence, it is based of a difference where you first identify the reduction potential in contrail avoidance, before considering other effects.

- **By this approach the percentages are the largest. If one would use SAF for targeted contrail regions first and then consider contrail avoidance, the results would be the opposite of your findings.**

This point has been addressed in response to the first of the “Major points” made by this reviewer. Equations 5, 6, 7 and 8 have been modified to explore the impact of contrail avoidance both with and without modified fuel composition. In addition, Figure 5 and paragraphs 7-9 of “The Impact of Different Contrail Avoidance Scenarios” have been modified to analyse and discuss this change.

page 17

the ECS is presented as unitless. Shouldn't it be in K?

The units of ECS have been corrected. The last two sentences of third paragraph of the “Aviation Climate Modelling” section of the Methods now read:

It yields an Equilibrium Climate Sensitivity (ECS) of 3.66 K (mean) and 5th to 95th percentile range of 1.94 K to 6.59 K. It also yields a Transient Climate Response (TCR) of 1.82 K (mean) and 5th to 95th percentile range of 1.30 K to 2.45 K.

References incomplete:

- **IEA. IEA, Oil Information. (2024).**
- **Klima, K. Assessment of a Global Contrail Modeling Method and Operational Strategies for Contrail Mitigation.**

These references have been corrected to:

- IEA. IEA, Oil Information. (2024) [doi:10.5257/iea/oil/2019-1](https://doi.org/10.5257/iea/oil/2019-1).
- Klima, K. Assessment of a Global Contrail Modeling Method and Operational Strategies for Contrail Mitigation.
[https://dspace.mit.edu/handle/1721.1/32460#:~:text=This%20thesis%20provides%20a%20model%20to%20assess%20operational,flight%20performance%20and%20best-available%20global%20meteorological%20data%20assimilations.\(2003\).](https://dspace.mit.edu/handle/1721.1/32460#:~:text=This%20thesis%20provides%20a%20model%20to%20assess%20operational,flight%20performance%20and%20best-available%20global%20meteorological%20data%20assimilations.(2003).)

Response to Reviewer 2

Thank you for the opportunity to review the paper "The Climate Opportunities and Risks of Contrail Avoidance,"

The paper evaluate the effect on global temperature change based on operationalization of a mitigation strategy: contrail avoidance.

This is a relevant topic. The methodology applied is appropriate. The results are reasonable.

The paper is well written.

We would like to thank Reviewer 1 for their positive and constructive feedback. We have made efforts to address each comment in point-by-point responses, as detailed below.

Section: The Impact of Different Contrail Avoidance scenarios.

The paper describes the impact of Different Start Dates (2035, 2040, 2045), Different Levels of Effectiveness, Modification of Fuel Composition

- 1. page 5-6, What is the exact definition of "Effectiveness"? What factors impact effectiveness? Does the ability to accurately identify ISS regions affect effectiveness?**

We have added some discussion on effectiveness and its limits in paragraph 4 of the introduction. It reads:

We here define "effectiveness" as the reduction in global contrail effective radiative forcing due to a fleet wide mitigation measure. Effective radiative forcing (ERF) is a measure of the climate forcing due to an emission after allowing for tropospheric temperature adjustments. At present, prediction of the geolocation and vertical extent of ISSRs is limited. This in turn imposes a limit on the effectiveness of contrail avoidance since a planned manoeuvre may be a) insufficiently large to avoid the full extent of the ISSR, b) unneeded since an ISSR was incorrectly predicted, or c) perverse since the manoeuvre deviates into an ISSR rather than away from it^{15,16}. There are other examples of factors that can impact the effectiveness of contrail avoidance, such as conflicts between the

flight paths of deviating and non-deviating aircraft and limitations on air traffic controller capacity⁸. It is possible to estimate effectiveness of contrail avoidance from flight trial and ISSR forecast data^{10,11,15,17} as in Smith et al. 2025¹⁸. However, there is high uncertainty in this estimate, and there are likely to be technological and operational advancements as further flight trials are conducted. It is therefore prudent to here examine a wide range of effectiveness values to fully explore the climate impacts of contrail avoidance.

- 2. Use of Matrices to visualize results. For Figures 2,3,4 would a matrix with cells of 2055 Temp Rise and columns and rows with the Factors make the results easier to summarize ? For example Figure 2 and 3: Row start years and Columns Effectiveness. Figure 4: Rows start year, columns Current Fuel, Modified Fuel.**

We have added Table 2 to include an alternative visualization of the results to Figure 3, Figure 4, and Figure 5 (equivalent to Figure 2, Figure 3 and Figure 4 in the original manuscript). The table contains every combination of reduction in global surface temperature rise in 2050 analysed in the figures, expressed in both Kelvin and as a percentage of the remaining temperature budget that is defined in “The No Action Scenario” section. The table cells are colour coded to indicate thresholds of reduction in global surface temperature rise.

Similarly, to provide an alternative visualisation to Figure 1, we have added Table 1 to the “Aviation Growth Projections Towards 2100”. For each of the aviation growth projections, the table details every combination of in global surface temperature rise in 2050 and 2100 analysed in Figure 1. The table cells are colour coded to indicate thresholds of global surface temperature rise.

- 3. Be clear which factor, Start Date, Effectiveness, or Fuel has the biggest influence?
If you had to pick one strategy, which would it be? Why?
More realistically, where would you recommend the research funding be spent: Start Date, Effectiveness, or Fuel?**

Table 2 allows the impacts of effectiveness, start date, and fuel modifications to be directly compared. In addition, we have added discussion directly comparing the changes in “The Impact of Different Contrail Avoidance Scenarios” section, including:

- To two significant figures, the mean reduction in global surface temperature due to contrail avoidance is seen to scale linearly with effectiveness. (*“The Impact of Different Contrail Avoidance Scenarios” paragraph 2*)
- Since the reduction in global surface temperature rise due to contrail avoidance in 2050 scales approximately linearly with effectiveness, the contrail avoidance scenario with 100% effectiveness and a start date of 2045 has approximately the same reduction in global surface temperature rise as the scenario with 22% effectiveness and a start date of 2035. It follows that a delay of 10 years is approximately equivalent to a 78% loss in effectiveness. (*“The Impact of Different Contrail Avoidance Scenarios” paragraph 4*)
- These observations highlight the importance of early contrail avoidance action, even if the operational and technological systems associated with the contrail avoidance strategy are not fully mature. Within realistic bounds, a contrail avoidance strategy with a start date of 2035 can lead to an earlier reduction, a lower peak, and a greater reduction in the global surface temperature rise than a strategy where action is delayed for the sake of higher technological readiness. Further, earlier contrail avoidance action provides more opportunities for learning as testing, development and the adoption of best practices occurs. This means that an earlier start date is likely to increase the effectiveness by 2050, amplifying the warming reduction that can be achieved by this date. (*“The Impact of Different Contrail Avoidance Scenarios” paragraph 6*)
- Therefore, a contrail avoidance scenario which is first implemented in 2035 is expected to achieve a greater contrail warming reduction than modified fuel compositions, provided the effectiveness of contrail avoidance is greater than approximately 32%. This assertion neglects any improvements in effectiveness that may occur as contrail avoidance is adopted throughout the fleet. In such cases, contrail avoidance achieves a greater contrail warming reduction than modified fuel compositions at effectiveness values that are <32%. (*“The Impact of Different Contrail Avoidance Scenarios” paragraph 9*).

4. Recommend explain why the Start Date is so important? It integrates over time!

Earlier contrail avoidance action means that there are more opportunities for learning as practices as testing, development and the adoption of best practices occurs. This means that an earlier start date is likely to increase the effectiveness by 2050, amplifying the warming reduction that can be achieved by this date. We have added some discussion on this and on the time integrated impact of start date in paragraphs 5 and 6 of “The Impact of Different Contrail Avoidance Scenarios” section:

Further, the global surface temperature rise due to contrails has a lower maximum value and declines earlier for earlier dates of first action. In the scenario where contrail avoidance is first introduced in 2035, the mean temperature peaks at 0.038 K in 2038. In the scenarios where it is first introduced in 2040 and 2045, the mean temperature peaks at 0.043 K in 2043 and 0.049 K in 2048, respectively. It follows that the reduction in global surface temperature rise integrated over time to 2050 (a measure the climate damage that has been prevented to 2050²) is higher in value for earlier start dates, equal to a mean of 0.325 K-year for 2035, 0.144 K-year for 2040, and 0.018 K-year for 2045.

These observations highlight the importance of early contrail avoidance action, even if the operational and technological systems associated with the contrail avoidance strategy are not fully mature. Within realistic bounds, a contrail avoidance strategy with a start date of 2035 can lead to an earlier reduction, a lower peak, and a greater reduction in the global surface temperature rise than a strategy where action is delayed for the sake of higher technological readiness. Further, earlier contrail avoidance action provides more opportunities for learning as testing, development and the adoption of best practices occurs. This means that an earlier start date is likely to increase the effectiveness by 2050, amplifying the warming reduction that can be achieved by this date.

- 5. Recommend a short discussion on the issue of accurately identifying the geographic and vertical location of Ice Super Saturated regions. See #1 above. What is current accuracy (e.g. from flight trials? What is current state-of-the-art in detection? How could this be improved? What about forecasting?**

We have addressed this in our response to point 1 above.

- 6. Methods Section: OK.**

We thank the reviewer for their positive feedback.

The Climate Opportunities and Risks of Contrail Avoidance – Response to Reviewers

We would like to thank the reviewers for their further feedback on our manuscript. This document provides a detailed response to the provided feedback. It summarises the changes we have made to improve the manuscript and includes direct references to the manuscript text. Some guiding notes for interpreting this document are detailed below:

- *The reviewer comments are detailed in **bold font**. Our point-by-point responses are detailed in non-bold, italicised font.*
- *Any direct quotes from the manuscript are in black, non-italicised text.*
- *Any inserted text is red, non-italicised and underlined.*
- *Any deleted text is not shown.*

Response to Reviewer 1

The authors have thoroughly revised their manuscript and adequately answered my questions. Thank you for efforts.

We in turn thank the reviewer for their constructive feedback which has helped us to improve the manuscript.

I have some minor, though important, comments that I would like the authors to consider:

- **I 27-28: The sentence gives the impression that the climate impact of aviation solely arises from CO₂ and contrails. Lee et al. 2021 gives a different view. I think it would be good to adapt this sentence, accordingly.**

To avoid the impression that the climate impact of aviation solely arises from CO₂ and contrails, we have reworded this sentence. Lines 26-30 now read:

If no avoidance is adopted, aviation is projected to contribute 0.040 K of CO₂ warming and 0.054 K of contrail warming by 2050. The combined warming from aviation CO₂ and contrails is 19% of the difference between current temperatures and the +2 °C limit above pre-Industrial levels, i.e. 19% of our remaining temperature budget.

- **I 42 You are referring to a temperature rise and give Lee et al. as a reference(3). However, I think that paper concentrates on RF and ERF rather than dT, correct?**

This is true, Lee et al. (2021) does focus on RF and ERF rather than temperature rise. We have adapted this sentence to refer instead to “warming” which is also consistent with the rest of the first paragraph. In addition, we now reference Aamaas et al. (2025), which does investigate global surface temperature rise due to contrails, as well as other non-CO₂ and CO₂ emissions from aviation. Lines 40-41 now read:

The current warming due to contrails is similar to that of all accumulated aviation CO₂ emissions since the beginning of the jet age^{3,4,5}.

Where ⁵ now refers to the following reference:

Aamaas, B. et al. Continued global warming from aviation even under high-ambition mitigation scenarios. One Earth 8, (2025).

The first reference to “global surface temperature rise” is now on line 48-49, which reads:

This study focuses on quantifying the size and speed of the reduction in global surface temperature rise that is achievable though contrail avoidance.

Aamaas et al. (2025) is also referred to in lines 93-95, which read:

Aamaas et al. 2025⁵ and Klöwer et al. 2021⁴ instead analyse the global surface temperature change due aviation CO₂ and non-CO₂ under various future aviation emission scenarios.

- **I 402 "Climate Modelling" in order to avoid confusion with GCM modelling I suggest to adapt to something like "simplified climate modelling".**

This sentence has been adapted to refer to simplified climate modelling. It now reads:

Reduced order climate modelling is used to assess the impact of contrail avoidance on global surface temperature change towards 2050.

- **Conclusion: As in the lines 64-68 the term "effectiveness" is also interpreted in terms of forecast abilities, it would be good to explicitly state again in the conclusion to what this term actually refers. To my understanding, the aspect of forecast abilities or worsening the situation is not covered.**

We have modified the first paragraph of the conclusion so that it now explicitly defines "effectiveness". Lines 323 to 332 now read:

Reduced order climate modelling is used to assess the impact of contrail avoidance on global surface temperature change towards 2050. We define "effectiveness" as the reduction in global contrail effective radiative forcing (ERF) due to a fleet-wide mitigation measure. Various contrail avoidance scenarios are investigated, including scenarios with different contrail effective radiative forcing (ERF) growth rates beyond 2045, levels of effectiveness of 0%, 25%, 50%, 75%, and 100%, dates of first action of 2035, 2040 and 2045, and levels of fleet averaged fuel burn penalties of 0.35%, 5% and 10% at full adoption. The contrail warming reduction impacts of contrail avoidance and modified fuel composition are also compared, and the combined impacts are explored.

Response to Reviewer 2

The Authors have responded to the Review comments satisfactorily. The paper is improved and easier to read. This reviewer recommends publication of the paper as-is.

We thank reviewer 2 for their feedback and for their recommendation.